# Prototype-based Aleatoric Uncertainty Quantification for Cross-modal Retrieval

**Hao Li**
18th.leolee@gmail.com

**Jingkuan Song**[*]
jingkuan.song@gmail.com

**Lianli Gao**
lianli.gao@uestc.edu.cn

**Xiaosu Zhu**
xiaosu.zhu@outlook.com

**Heng Tao Shen**
shenhengtao@hotmail.com

Center for Future Media,
University of Electronic Science and Technology of China

## Abstract

Cross-modal Retrieval methods build similarity relations between vision and language modalities by jointly learning a common representation space. However, the predictions are often unreliable due to the *Aleatoric* uncertainty, which is induced by low-quality data, *e.g*., corrupt images, fast-paced videos, and non-detailed texts. In this paper, we propose a novel Prototype-based Aleatoric Uncertainty Quantification (PAU) framework to provide trustworthy predictions by quantifying the uncertainty arisen from the inherent data ambiguity. Concretely, we first construct a set of various learnable prototypes for each modality to represent the entire semantics subspace. Then *Dempster-Shafer Theory* and *Subjective Logic Theory* are utilized to build an evidential theoretical framework by associating evidence with Dirichlet Distribution parameters. The PAU model induces accurate uncertainty and reliable predictions for cross-modal retrieval. Extensive experiments are performed on four major benchmark datasets of MSR-VTT, MSVD, DiDeMo, and MS-COCO, demonstrating the effectiveness of our method. The code is accessible at https://github.com/leolee99/PAU.

## 1 Introduction

Cross-modal Retrieval, devoted to searching a gallery of related samples in vision/language modality given a query in another, has attracted increasing interest as the wellspring development of multi-media platforms. However, the heterogeneity between modalities casts daunting challenges for this task. To tackle these challenges, most existing methods [9, 60, 19, 15, 4, 53, 49] map the query and retrieval items from distinct modalities into a joint latent space to make similarity calculation feasible.

Some early works [9, 60] corroborate the powerful capabilities of deep neural networks to extract high-level features. Chen *et al*. [9] construct a hierarchical graph structure for videos and texts to build the relationships between the local and the global features. PVSE [60] utilizes the multi-head self-attention module to learn multiple and diverse representations of videos and texts for the polysemous problem. Recently, a large body of transformer-based pre-training models [19, 15, 4, 53, 49] have shown remarkable superiority with excellent generalization and performance.

However, prior approaches routinely assume the data qualities are absolutely stable and view them equally. They estimate the relevance of vision-language pairs relying only on the similarity scores

---

[*]Corresponding author.

37th Conference on Neural Information Processing Systems (NeurIPS 2023).

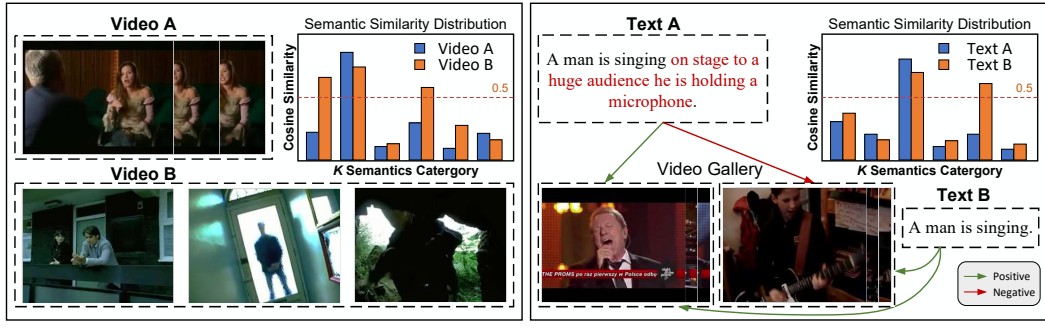

(a) Video modality                                        (b) Text modality

Figure 1: **Illustration of confused matching in fast-paced videos and non-detailed texts.** Assuming the possible semantics of each modal subspace are finite with $K$ categories. (a) A single-scene *Video A* can only match one semantics of "*talking*". By contrast, a multi-scene *Video B* can match to 3 semantics of "*talking*", "*shadow*", and "*cave*". (b) *Text A* can only match the left video, while *Text B* with some details removed (in red) matches both videos.

generated by complex neural networks, but ignoring the confidence of these predictions, which should be taken seriously. Low-quality data like corrupt images [31], fast-paced videos [26], non-detailed sentences [34], *etc.*, inevitably lead to unreliable results. The model therefore not only needs to predict the relevance of vision-language pairs, but also capably answer the question "*How confident is the prediction?*". Namely, it is necessary to quantify the uncertainty of each data to guide more reliable vision-language pair similarity.

In the literature, uncertainty can be classified into *Aleatoric Uncertainty* and *Epistemic Uncertainty* based on causation [32]. The former is induced by the inherent data defects, while the latter is inherent to the model. This work mainly focuses on the aleatoric uncertainty (*i.e.*, data uncertainty), defined as "the uncertainty caused by ambiguous multi-modal data, such as fast-paced videos and non-detailed texts". Next, we will answer the question "*Why do these ambiguous data lead to uncertainty?*". We first assume the possible semantics of each modal subspace are finite with $K$ categories. Afterward, the fast-paced videos (multi-scene videos) and the non-detailed texts are generally highly related to multiple semantics, misleading models into confused matching, shown in the Figure 1. These alternative matches will bring uncertainty, which can be explained through *Information Entropy* [59]. Concretely, Information Entropy is the average uncertainty level in the random variable's possible outcomes, which can be denoted as Eq. 1 with the possible semantics as the random variable $X$.

$$\mathcal{H}(X) = \mathbb{E}[-\log p(x_i)] = -\sum_{i=1}^{K} p(x_i) \log p(x_i), \qquad (1)$$

where $\sum_{i=1}^{K} p(x_i) = 1$, $p(x_i)$ is the $i^{th}$ semantics probability derived from semantic cosine similarity through a softmax layer. The ambiguous data highly associated with multiple semantics have a low variance of the probability distribution, resulting in high information entropy and high uncertainty due to "*The Maximum Entropy Principle* [33]" (proof in Appendix. E). Information entropy is a seemingly reasonable criterion to estimate the data uncertainty, but unfortunately, it is powerless to distinguish the instances with the similarity distribution of $[0.8, 0.8, 0.8, 0.8]$ and the distribution of $[0.2, 0.2, 0.2, 0.2]$. After a softmax function, both distributions turn into $[0.25, 0.25, 0.25, 0.25]$ and obtain the same information entropy. In practice, we hope the first one gets a higher uncertainty score than the other because the front of the retrieval sequence with highly similar instances is more notable for the retrieval task. Thus, the *Dempster-Shafer Theory of Evidence* (DST) [58], a generalization of the Bayesian theory to subjective probabilities, which has developed into a general framework for uncertainty modeling, is leveraged to quantify multi-modal aleatoric uncertainty.

To precisely quantify the aleatoric uncertainty, in this work, we propose an effective Prototype-based Aleatoric Uncertainty Quantification (PAU) framework to provide trustworthy predictions by quantifying the uncertainty arisen from inherent data ambiguity. To be specific, we first construct a series of diverse prototypes for each modality to capture the overall $K$ semantics of the subspace. Each prototype represents an individual semantics. Afterward, the variations of cosine similarities

between the instance and the prototypes from another modality are deployed as the belief masses used in DST [59] to model uncertainty. Finally, a re-ranking practice is utilized to further strengthen prediction reliability by regarding each data uncertainty as a similarity weight.

The main contributions of our paper are three folds. 1) We give a clear definition of the inherent aleatoric uncertainty in multi-modal data. 2) A powerful aleatoric uncertainty quantification framework PAU is built to accurately estimate the data uncertainty and effectively mitigate the negative impact on retrieval. 3) We also verify the superiority of our proposed PAU on multiple challenging benchmarks. Massive insightful ablation studies further reveal its effectiveness and generalization.

## 2 Related work

### 2.1 Cross-modal Retrieval

Cross-modal retrieval [9, 25, 67, 39, 17, 60, 54, 49, 22, 18, 66, 26, 64, 37], as one of the most popular tasks in multi-modal learning, has attracted increasing attention with the aim of searching similar semantic samples from different modalities. Early works [25, 39] build a joint embedding space to make query-positive pairs closer by canonical correlation analysis (CCA) [28]. Faghri *et al*. [17] first use a triplet loss to improve the model performance by focusing only on the hardest negative. PVSE [60] proposes a multi-candidate representation function to capture the diversity and build one-to-many relationships in different modalities. Recently, CLIP [54] has shown vital power and generalization in various multi-modal tasks. A rich body of CLIP-based approaches [49, 22, 18, 66, 26] have got significant achievements in cross-modal retrieval. CLIP4Clip [49] transfers the rich knowledge from image-text pre-trained model CLIP to video-text retrieval. CenterCLIP [66] proposes a multi-segment token clustering algorithm to keep the most representative tokens in consecutive frames. In this paper, we follow the setting of the above work, which inherits the knowledge from CLIP.

### 2.2 Uncertainty Quantification

Generally, there are two paradigms among existing uncertainty estimation approaches. The first one is the Bayesian-based paradigm [5, 51, 40], which estimates uncertainty by approximating the moments of the posterior predictive distribution. Subsequently, some algorithms focus on the variety of Bayesian such as Laplace approximation [50], Markov Chain Monte Carlo [52], and variational techniques [55, 6]. Recent work [21] leverages dropout [61] in the test phase, reducing the computational cost of uncertainty estimation. Nonetheless, limited by the increment of model parameters and poor convergence, these approaches always have expensive training costs and slow inference time. The second paradigm focuses on the non-Bayesian approach [41, 57, 27]. Lakshminarayanan *et al*. [41] propose an alternative to Bayesian Neural Networks (BNN) which is simple to train and produce reliable uncertainty estimates. Wang *et al*. [57] replace the parameter set of a categorical distribution with the parameters of a Dirichlet density and represent the uncertainties with a single abnormal class. Recently, uncertainty-based algorithms have been used in multi-view classification for trusted decision-making [27]. Although uncertainty-based methods have achieved impressive progress, they are limited to working with particular classification and segmentation tasks.

## 3 Method

In this section, we present our **Prototype-based Aleatoric Uncertainty Quantification (PAU)** framework (Figure 2). We first define the cross-modal retrieval task and introduce a commonly used pipeline in Section 3.1. Next, we detail the uncertainty theory, as well as the approach to quantify the uncertainty in Section 3.2.

### 3.1 Task Definition

Let $\mathcal{D} = (\mathcal{V}, \mathcal{T})$ denote a vision and language dataset, where $\mathcal{V}$ is a set of images or videos, and $\mathcal{T}$ is a set of texts. The goal of cross-modal retrieval is to rank the relevance between a visual query $v \in \mathcal{V}$ (respectively a textual query $t \in \mathcal{T}$) and textual set $\mathcal{T}$ (respectively visual set $\mathcal{V}$). Recent works [54, 49, 22, 18, 66, 26] have shown CLIP's strong performance and generalization in various downstream tasks, inspiring us to employ CLIP as our backbone.

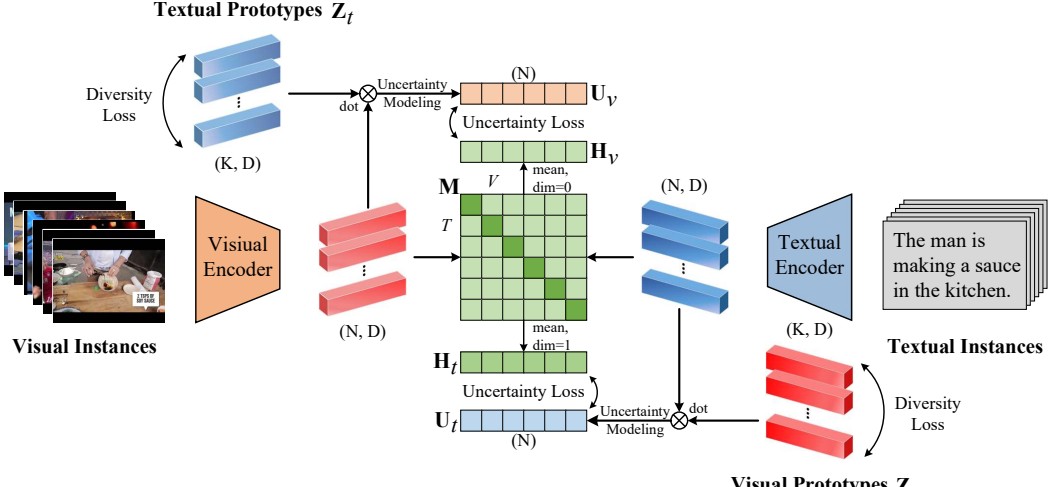

Figure 2: **The Framework of PAU.** The visual encoder $\phi_v$ and textual encoder $\phi_t$ separately map the visual and textual instances into a joint embedding space to calculate the similarity matrix **M**. A dot product function is used to build a set of similarity vector $\mathbf{P} \in \mathbb{R}^{N \times K}$ between $N$ instances and $K$ prototypes, afterward modeling the uncertainty. An uncertainty loss forces the prototypes into learning the rich semantics of subspace to realize accurate uncertainty quantification. Besides, A diversity loss is introduced to keep prototypes diverse. $\otimes$ means cosine similarity.

To be specific, given a visual instance $v$ and a text instance $t$, the CLIP encodes them into a latent embedding space $\mathbb{R}^D$ through a visual encoder $\phi_v(\cdot)$ and a textual encoder $\phi_t(\cdot)$, respectively. Then the relevance between them can be computed using a cosine similarity function as:

$$\text{sims}(v, t) = \cos(\phi_v(v), \phi_t(t)) = \frac{\phi_v(v)}{\|\phi_v(v)\|} \cdot \frac{\phi_t(t)}{\|\phi_t(t)\|} \tag{2}$$

### 3.2 Uncertainty Quantification

**Prototype construction.** Although the cosine similarity function has shown effectiveness in most existing works [25, 60, 54, 49], some unreliable results will inevitably be produced due to the low inherent data quality. These ambiguous data are generally highly related to multiple semantics, leading to confusing matching (see Figure 1).

To assess these data uncertainty and provide trustworthy predictions, we first assume the semantic space is finite with $K$ semantic categories. Then, $K$ learnable prototypes $\mathbf{Z} \in \mathbb{R}^{K \times D}$ are constructed to represent the overall latent semantic subspace for each modality, making data semantic ambiguity evaluable. Each prototype represents an individual semantics. Finally, DST [12] is employed to build the uncertainty quantification framework by assigning the variations of similarities between an instance and the prototypes from another modality as the belief masses.

**The theory of uncertainty.** The *Dempster-Shafer Theory of Evidence* (DST) [12], a generalization of the Bayesian theory to subjective probabilities [12], builds a general uncertainty modeling framework by assigning belief masses to the set of mutually exclusive possible states with one state expressing "*I am not sure*". Following EDL [57], we adopt *Subjective Logic* (SL) [35] to associate the belief distribution in DST with the parameters of the Dirichlet distribution. More precisely, SL provides an evidential theoretical framework to quantify the belief masses $b_k$ of different categories and overall uncertainty mass $\psi$ for the $K$ classification task. These $K + 1$ mass values are all non-negative and sum up to one:

$$\psi + \sum_{k=1}^{K} b_k = 1, \tag{3}$$

where $b_k \geq 0$ and $\psi \geq 0$ denote the belief mass of the $k^{th}$ semantic category ($k = 1, \cdots, K$) and overall uncertainty mass, respectively. In our setting, $\psi$ here means the uncertainty mass of the

event "the data is ambiguous", which is negatively related to the degree of data ambiguity. Thus, the $u = 1 - \psi = \sum_{k=1}^{K} b_k$ is designed to express the aleatoric uncertainty defined in Section 1. In brief, the data with lower $\psi$ is more ambiguous, which should be a high aleatoric uncertainty score as higher $u$. A belief mass $b_k$ for the $k^{th}$ semantic category is computed using the evidence for the category. We term the *evidence* as the amount of data supports for being classified into a certain category. Let $\mathbf{e} = [e_1, \cdots, e_k]$ be an evidence vector derived from similarity vector $\mathbf{p} = [p_1, \cdots, p_k]$ between an instance and $K$ prototypes (evidence generation shown in Appendix. B), then the belief $b_k$ and the uncertainty $u$ are signified as:

$$b_k = \frac{e_k}{S} = \frac{\alpha_k - 1}{S} \quad \text{and} \quad u = 1 - \psi = \sum_{k=1}^{K} b_k = 1 - \frac{K}{S}, \tag{4}$$

where $\alpha_k = e_k + 1$ denote the Dirichlet parameters associated with a belief mass assignment in SL, and $S = \sum_{i=1}^{K} \alpha_k$ is referred to as the Dirichlet strength. $e_k$ is the evidence of a visual (or textual) instance matching the $k^{th}$ semantics category. Intuitively, massive evidence indicates considerably ambiguous data, resulting in high uncertainty, which brings out fallible predictions.

**Uncertainty learning.** After uncertainty modeling, how to force the model to learn uncertainty from data is worth considering (see Figure 2). The key is enriching prototypes' meaningful semantics to represent overall subspace rigorously. We first give a normalized initialization proposed by Xavier [24] to all prototypes. Subsequently, the cosine similarity matrix $\mathbf{M} \in \mathbb{R}^{N \times N}$ between $N$ visual and textual instances is calculated by Eq. 2 within a batch. Furthermore, $N$ instances and $K$ prototypes from different modalities build a set of similarity vectors $\mathbf{P} \in \mathbb{R}^{N \times K}$ to derive the evidence vector set and compute the uncertainty using Eq. 4. The data uncertainty sets of $N$ visual and textual instances can be symbolized as $\mathbf{U}_v = [u_1^v, \cdots, u_N^v]$ and $\mathbf{U}_t = [u_1^t, \cdots, u_N^t]$ in parallel.

Next, an objective function is required to force prototypes to learn rich semantic knowledge. As the prototypes are expected to represent the overall semantics of each modality, the instance with high overall similarity to these prototypes should also have high overall similarity with other instances from another modality. According to Eq. 4, the aleatoric uncertainty $u$, denoted as the sum of belief masses, should be positively related to the mean similarities of each modality within a batch. Thereby, we compute the mean of similarity matrix $\mathbf{M}$ along the row and the column, symbolized as $\mathbf{H}_v = [h_1^v, \cdots, h_N^v]$ and $\mathbf{H}_t = [h_1^t, \cdots, h_N^t]$, respectively. After that, prototypes can be optimized by employing Mean-Squared (MSE) Loss to keep the $\mathbf{U}_v$ ($\mathbf{U}_t$) and $\mathbf{H}_v$ ($\mathbf{H}_t$) consistent, where $\lambda$ is a scaling coefficient to keep the same value range of $u$ and $h$ (Eq. 5).

$$\mathcal{L}_{uct}^v = \frac{1}{N} \sum_{i=1}^{N} (u_i^v - \lambda \cdot h_i^v)^2$$

$$\mathcal{L}_{uct}^t = \frac{1}{N} \sum_{i=1}^{N} (u_i^t - \lambda \cdot h_i^t)^2 \tag{5}$$

**Diversity learning.** To employ the DST uncertainty framework, the mutual exclusion among possible states needs to be satisfied. Precisely, prototypes should avoid attending to the same or similar semantics. Therefore, a diversity loss is proposed to help the prototypes focus on diversified independent semantics. The formula is as:

$$\mathcal{L}_{div}^v = \frac{1}{K^2} \sum_{i=1}^{K} \sum_{j=1}^{K} (\cos(\mathbf{z}_i^v, \mathbf{z}_j^v))^2$$

$$\mathcal{L}_{div}^t = \frac{1}{K^2} \sum_{i=1}^{K} \sum_{j=1}^{K} (\cos(\mathbf{z}_i^t, \mathbf{z}_j^t))^2 \tag{6}$$

where $\mathbf{z}_i^v$ and $\mathbf{z}_i^t$ denote $i^{th}$ prototypes of visual and textual modality. Intuitively, the semantic similarities between different prototypes are expected to be 0.

**Re-ranking.** After training, the uncertainty of each instance could be precisely assessed using Eq. 4. To further alleviate the impact of the uncertainty on prediction, we re-rank the retrieval sequences by weighting the predicted matrix based on the uncertainty vector $\mathbf{U}_v$ and $\mathbf{U}_t$ by Eq. 7.

$$\mathbf{M}'' = (\exp(-\beta_1 \mathbf{U}_v^T) \cdot \exp(-\beta_2 \mathbf{U}_t)) \circ \mathbf{M}' \tag{7}$$

where $\beta_1$ and $\beta_2$ are learnable parameters used to control the impact degree of re-ranking on prediction. $\mathbf{M}'$ and $\mathbf{M}''$ separately signify the similarity matrices between all visual and textual instances in test processing before and after re-ranking.

# 4 Experiments

In this section, we first introduce the datasets and metrics used for our experiments, as well as the implementation details. Later, we compare PAU with the previous state of the arts on several tasks and public benchmarks. In the end, rich ablation studies are carried out to verify our method's effectiveness and generalization.

## 4.1 Datasets and Metrics

We report our experimental results on three public video-text retrieval datasets containing MSR-VTT [62], MSVD [7], DiDeMo [30], and a public image-text retrieval dataset of MS-COCO [45]. Following [49], Recall@1 (R@1), Recall@5 (R@5), Recall@10 (R@10), Median Rank (MdR), and Mean Rank (MnR) are used to evaluate the performance.

**MSR-VTT** [62] is a commonly used benchmark in video-text retrieval, containing 10,000 Youtube videos with 20 captions for each. Following [20], we employ 9,000 videos for training and 1,000 selected video-text pairs for testing.

**MSVD** [7] contains 1,970 videos ranging in length from 1 to 62 seconds, with approximately 40 sentences per video. There are 1,200, 100, and 670 videos used for training, validation, and testing in the standard split.

**DiDeMo** [30] consists of 10,611 Flickr videos with approximately 40,000 annotations. We follow [46, 42, 3] and concatenate all captions of the same video into a paragraph, which will be regarded as a single item in a paragraph-to-video retrieval task.

**MS-COCO** [45] is a widely used dataset for image-text retrieval, containing 123,287 images with five captions per image. We follow the split in [36] with 113,287 images for training, 5,000 for validation, and 5,000 for testing. The results are reported on 1K and 5K test sets.

Table 1: Retrieval Performance on MSR-VTT

| Method | t2v | | | | | v2t | | | | |
|---|---|---|---|---|---|---|---|---|---|---|
| | R@1 | R@5 | R@10 | MdR | MnR | R@1 | R@5 | R@10 | MdR | MnR |
| CLIP4Clip-meanP [49] | 43.1 | 70.4 | 80.8 | 2.0 | 16.2 | 43.1 | 70.5 | 81.2 | 2.0 | 12.4 |
| CLIP4Clip-seqTransf [49] | 44.5 | 71.4 | 81.6 | 2.0 | 15.3 | 42.7 | 70.9 | 80.6 | 2.0 | 11.6 |
| CenterCLIP [66] | 44.2 | 71.6 | 82.1 | 2.0 | 15.1 | 42.8 | 71.7 | 82.2 | 2.0 | 10.9 |
| MILES [23] | 44.3 | 71.1 | 80.2 | 2.0 | 14.7 | - | - | - | - | - |
| CLIP2Video [18] | 45.6 | 72.6 | 81.7 | 2.0 | 14.6 | 43.5 | 72.3 | 82.1 | 2.0 | 10.2 |
| CLIP2TV [22] | 46.1 | 72.5 | **82.9** | 2.0 | 15.2 | 43.9 | 73.0 | 82.8 | 2.0 | 11.1 |
| XPool [26] | 46.9 | 72.8 | 82.2 | 2.0 | 14.3 | 44.4 | **73.3** | **84.0** | 2.0 | **9.0** |
| **Baseline** | 45.6 | **72.9** | 81.5 | 2.0 | 14.5 | 45.2 | 71.6 | 81.5 | 2.0 | 10.9 |
| **ours** | **48.5** | 72.7 | 82.5 | **2.0** | **14.0** | **48.3** | 73.0 | 83.2 | **2.0** | 9.7 |

Table 2: Retrieval Performance on MSVD

| Method | t2v | | | | | v2t | | | | |
|---|---|---|---|---|---|---|---|---|---|---|
| | R@1 | R@5 | R@10 | MdR | MnR | R@1 | R@5 | R@10 | MdR | MnR |
| CLIP4Clip-meanP [49] | 46.2 | 76.1 | 84.6 | 2.0 | 10.0 | 56.6 | 79.7 | 84.3 | 1.0 | 7.6 |
| CLIP4Clip-seqTransf [49] | 45.2 | 75.5 | 84.3 | 2.0 | 10.0 | 62.0 | 87.3 | 92.6 | 1.0 | 4.3 |
| CenterCLIP [66] | 47.3 | 76.9 | **86.0** | 2.0 | 9.7 | 63.5 | 86.4 | 92.6 | 1.0 | 3.8 |
| CLIP2Video [18] | 47.0 | 76.8 | 85.9 | 2.0 | 9.6 | 58.7 | 85.6 | 91.6 | 1.0 | 4.3 |
| CLIP2TV [22] | 47.0 | 76.5 | 85.1 | 2.0 | 10.1 | - | - | - | - | - |
| XPool [22] | 47.2 | 77.4 | 86.0 | 2.0 | **9.3** | 66.4 | 90.0 | 94.2 | 1.0 | 3.3 |
| **Baseline** | 46.3 | 76.8 | 84.9 | 2.0 | 10.0 | 60.1 | 82.8 | 87.3 | 1.0 | 7.5 |
| **ours** | **47.3** | **77.4** | 85.5 | **2.0** | 9.6 | **68.9** | **93.1** | **97.1** | **1.0** | **2.4** |

Table 3: Retrieval Performance on DiDeMo

| Method | t2v R@1 | R@5 | R@10 | MdR | MnR | v2t R@1 | R@5 | R@10 | MdR | MnR |
|---|---|---|---|---|---|---|---|---|---|---|
| CLIP4Clip-meanP [49] | 43.4 | 70.2 | 80.6 | 2.0 | 17.5 | 43.4 | 69.9 | 80.2 | 2.0 | 17.5 |
| CLIP4Clip-seqTransf [49] | 43.4 | 69.9 | 80.2 | 2.0 | 17.5 | 42.7 | 70.9 | 80.6 | 2.0 | 11.6 |
| CLIP2TV [22] | 45.5 | 69.7 | 80.6 | 2.0 | 17.1 | - | - | - | - | - |
| **Baseline** | 47.0 | 76.0 | **86.1** | 2.0 | **10.7** | 46.9 | **74.4** | 84.2 | 2.0 | **8.4** |
| **ours** | **48.6** | 76.0 | 84.5 | **2.0** | 12.9 | **48.1** | 74.2 | **85.7** | 2.0 | 9.8 |

Table 4: Comparison on MS-COCO

| Method | Backbone | MS-COCO 1K i2t R@1 | R@5 | R@10 | t2i R@1 | R@5 | R@10 | MS-COCO 5K i2t R@1 | R@5 | R@10 | t2i R@1 | R@5 | R@10 |
|---|---|---|---|---|---|---|---|---|---|---|---|---|---|
| PVSE [60] | | 69.2 | 91.6 | 96.6 | 55.2 | 86.5 | 93.7 | 45.2 | 74.3 | 84.5 | 32.4 | 63.0 | 75.0 |
| SGRAF [13] | ResNet-101 | 79.6 | 96.2 | 98.5 | 63.2 | 90.7 | 96.1 | 57.8 | - | 91.6 | 41.9 | - | 81.3 |
| NAAF [65] | | 80.5 | 96.5 | 98.8 | 64.1 | 90.7 | 96.5 | 58.9 | 85.2 | 92.0 | 42.5 | 70.9 | 81.4 |
| DAA [43] | | 80.2 | 96.4 | 98.8 | 65.0 | 90.7 | 95.8 | 60.0 | 86.4 | 92.4 | 43.5 | 72.3 | 82.5 |
| VSE∞ [8] | | **82.0** | **97.2** | 98.9 | 69.0 | 92.6 | 96.8 | 62.3 | **87.1** | **93.3** | **48.2** | **76.7** | **85.5** |
| PCME [11] | ViT-B/32 | 80.1 | 96.6 | 98.7 | 67.6 | 92.1 | 96.9 | 59.9 | 85.8 | 92.3 | 46.1 | 75.0 | 84.6 |
| PCME++ [10] | | 81.6 | **97.2** | **99.0** | **69.2** | **92.8** | **97.1** | 62.1 | 86.8 | 93.3 | 48.1 | 76.7 | 85.5 |
| **Baseline** | | 80.1 | 95.7 | 98.2 | 67.1 | 91.4 | 96.6 | 62.9 | 84.9 | 91.6 | 46.5 | 73.8 | 82.9 |
| **ours** | | 80.4 | 96.2 | 98.5 | 67.7 | 91.8 | 96.6 | **63.6** | 85.2 | 92.2 | 46.8 | 74.4 | 83.7 |

## 4.2 Implementation Details

We use CLIP's [54] visual-textual encoder (ViT-B/32) as the backbone and initialize all parameters from pre-trained CLIP weights. The max token and frame length are both set consistently with [49]. Specifically, the dimension of common embedding space is set to 512. In video-text retrieval, the Adam optimizer [38] is employed with the initial learning rate of 1e-7 for the visual encoder and the textual encoder, as well as the initial learning rate of 1e-4 for other layers. As for image-text retrieval, the AdamW optimizer [48] updates the parameters of all modules with the initial learning rate of 1e-5. The learning rate decays complying with the cosine schedule strategy [47] in 5, 3, 20, and 5 epochs trained on MSR-VTT, MSVD, DiDeMo, and MS-COCO. Besides, the number of prototypes $K$ is set as 8 according to Table 8. All experiments are conducted on 1 to 4 RTX3090.

## 4.3 Comparison with State of the Arts

**Video-text retrieval.** We compare our model with other state-of-the-art methods [49, 66, 18, 22, 26, 23, 54] and our baseline (model structure see Appendix. A) on multiple datasets. For fair comparisons, all methods are CLIP-based, symbolizing the most advanced techniques of video-text retrieval. The performance of baseline trained without PAU is also given. Table 1, 2, 3 show the results of video-text retrieval on MSR-VTT, MSVD, and DiDeMo, respectively.

We can observe that PAU brings sizeable improvements on all benchmarks. For text-to-video retrieval (t2v), PAU boosts the R@1 performance of 2.9%, 1.0%, and 1.6% on MSR-VTT, MSVD, and DiDeMo. Moreover, 3.1%, 8.8%, and 1.2% improvements of R@1 are obtained on video-to-text retrieval (v2t). Notably, we find that the performance improvement of v2t on MSVD is more significant than other benchmarks. The reason is that MSVD is a small dataset with few videos (only 1,200 for training) but massive annotations (about 40 annotations per video). Meanwhile, CLIP simply utilizes cross-entropy loss, which only regards the diagonal pairs as the positives. Nonetheless, for a batch with a typical size of 256 [49], the probability of any two texts coming from separate videos is approximately $3.49 \times 10^{-13}$ (see Appendix. D), bringing innumerable noises and performance damage for the training on video-to-text retrieval. Thus, the significant improvement indicates the ability of PAU to capture and rectify misleading data to support robust learning.

**Image-text retrieval.** We also test our method on image-text retrieval task. To be specific, CLIP [54] is utilized as the baseline. We compare our model with several previous state-of-the-art approaches, including PVSE [60], SGRAF [13], NAAF [65], DAA [43], VSE∞ [8], PCME [11], and PCME++ [10]. Table 4 shows the results of image-text retrieval on MS-COCO 1K and MS-COCO 5K, respectively.

We can observe that the overall performance improvement of image-text retrieval is smaller than that of video-text retrieval. The phenomenon reflects that video information is more complex than image information, inducing immense uncertainty in video-text retrieval. Furthermore, we also validate the robustness of PAU training in noisy datasets. The detailed results can be found in Appendix. C.4.

## 4.4 Comparison with Probabilistic Model

PCME [11] is a probabilistic-based approach to represent instance diversity, which can also represent uncertainty. The more diverse the instance, the more uncertain the instance. Thus, an instance's uncertainty and diversity should be positively correlated. To verify the superiority of our SL-based method over than probabilistic-based method, we conduct two experiments, *pearson correlation analysis* and *uncertain data analysis*.

**Pearson correlation analysis.** The pearson correlation coefficient (**PCC**) $r_{xy}$ ($r_{xy} \in [-1, 1]$) is a metric to estimate the strength of the correlation between variable $x$ and variable $y$. Generally, $r_{xy} > 0.5$ means that two variables are strongly positively correlated. As mentioned before, the uncertainty and the diversity of an instance should be positively correlated for PCME. If an instance is diverse, it will have high mean

Table 5: PCCs between $u$ and $h$

| Method | $PCC_V$ | $PCC_T$ | Dataset |
|---|---|---|---|
| PCME | -0.071 | -0.219 | COCO |
| PAU | 0.886 | 0.786 | COCO |
| PAU | 0.917 | 0.939 | MSR-VTT |

similarity $h$ with the instances from the other modality. This means that the uncertainty score $u$ and mean similarity $h$ should also be positively correlated for an instance. Thus, we compute the pearson correlation coefficient between $u$ and $h$ to explore whether PAU correctly estimates uncertainty. The results are shown in Table 5.

In this Table, we compute the PCCs of PCME and PAU on MSCOCO, as well as PCCs of PAU on MSR-VTT. $PCC_V$ and $PCC_T$ separately indicate the instance's PCC in vision or textual modality. The low PCCs of PCME mean current probabilistic models have limited abilities to reflect uncertainty and diversity. We argue the reason arises from the strong prior distribution assumption, *i.e.*, gaussian or other simple distributions are powerless to fit complex and diverse relationships in high-dimension space. By contrast, PAU obtains high PCCs in all settings, proving our approach indeed leads to proper uncertainty estimations.

**Uncertain data analysis**. To further prove the superiority of SL-based aleatoric uncertainty estimation method than probabilistic-based method, we compare the changes of R@1 after removing top-r instances with highest uncertainty scores predicted by PCME and PAU, respectively. To fairly compare, we employ removal on both predictions arised from PAU and PCME. The results with removed ratio from 0% to 20% on MS-COCO 5K is shown in Figure 3. We can observe that PAU outperforms PCME on all directions and predictions. This means that the uncertain data found by PAU is more precise than PCME. All experiments indicate the SL-based approach (PAU) works better than probabilistic-based method (PCME) on representing aleatoric uncertainty.

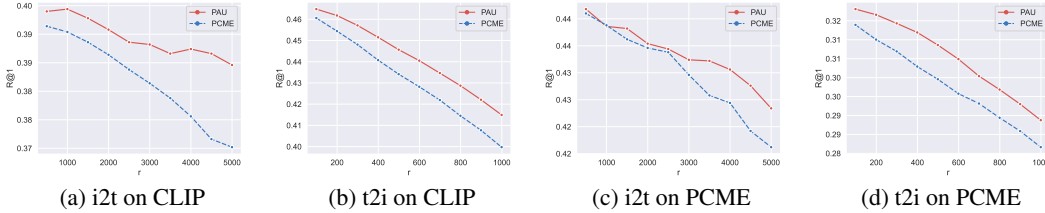

(a) i2t on CLIP      (b) t2i on CLIP      (c) i2t on PCME      (d) t2i on PCME

Figure 3: **The performance changes comparison after removing top-r instances with the highest uncertainty scores quantified by PCME and PAU on MS-COCO.** To fairly compare, we employ the removal on both predictions arising from CLIP and PCME. (a) and (b) show the performance changes on CLIP predictions. (c) and (d) show the performance changes on PCME predictions. In i2t, text instances are removed. In t2i, image instances are removed.

Table 6: Ablation study to investigate the effect of different components on MSR-VTT

| No | w/ $\mathcal{L}_{uct}$ | w/ $\mathcal{L}_{div}$ | Re-rank | t2v R@1 | R@5 | R@10 | v2t R@1 | R@5 | R@10 |
|----|----|----|----|----|----|----|----|----|----|
| 1 | | | | 45.6 | 72.9 | 81.5 | 45.2 | 71.6 | 81.5 |
| 2 | | | ✓ | 45.9 | 72.8 | 81.5 | 46.0 | 72.4 | 81.7 |
| 3 | ✓ | | | 47.0 | 72.4 | 82.8 | 47.4 | 71.8 | 82.2 |
| 4 | ✓ | | ✓ | 47.8 | 72.7 | 82.5 | 47.9 | 72.2 | 83.0 |
| 5 | ✓ | ✓ | | 48.4 | 72.4 | 82.2 | 47.7 | 72.6 | 82.6 |
| 6 | ✓ | ✓ | ✓ | 48.5 | 72.7 | 82.5 | 48.3 | 73.0 | 83.2 |

## 4.5 Ablation Studies

**The effect of different components.** To understand the effect of each component, we exhaustively ablate three main components of PAU, including Uncertainty Loss, Diversity Loss, and Re-ranking. As shown in Table 6, all three components effectively improve the model performance. The comparison between No. 1 and No. 3 shows that uncertainty loss gives significant R@1 improvements of 1.4% on t2v and 2.2% on v2t. The experiment No. 5 outperforms the experiment No. 3 by a clear margin, demonstrating the necessity of diversity loss. Furthermore, the experiment No. 2 re-ranks the baseline prediction using uncertainty vectors from PAU. It's worth noting that R@1 performance still boosts on both t2v and v2t, reflecting the strong generalization of PAU.

**The impact of uncertain data.** To demonstrate the uncertain data found by PAU indeed affect the retrieval, a trial of the retrieval subsets with varying reliability levels is carried out on MSR-VTT. Specifically, the samples are first ranked according to the uncertainty calculated by PAU in descending order. Then, $r$ samples with the highest uncertainty will be removed from the test set to produce an uncertainty-based removal subset. Meanwhile, $r$ random samples will be removed from the test set to produce a random-based removal subset. The baseline model is tested on both subsets above and reports the results in Figure 4.

It's worth noting that the model tested on the subsets removing uncertain data always outperforms the ones removing randomly. Furthermore, the performance gap reaches a peak of 14.2 when $r = 600$

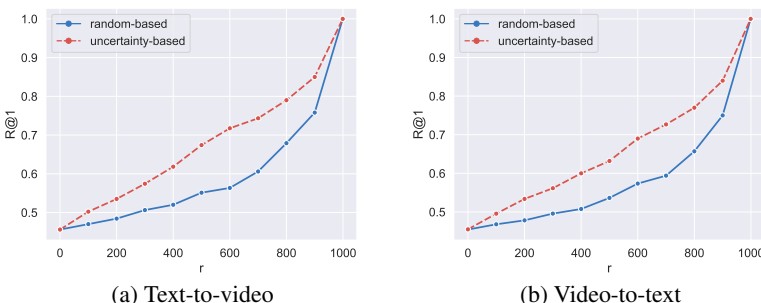

(a) Text-to-video      (b) Video-to-text

Figure 4: **The performance comparison against data removal number on MSR-VTT.**

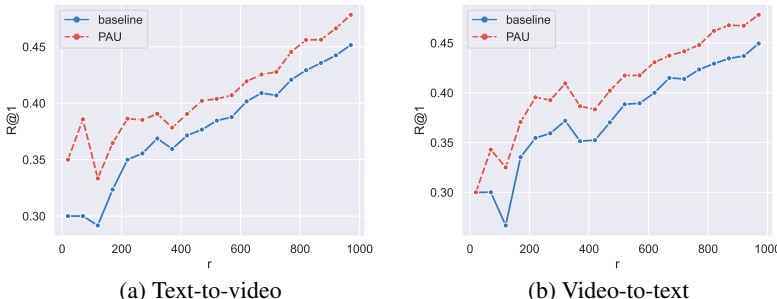

(a) Text-to-video      (b) Video-to-text

Figure 5: **The performance comparison against uncertain data number on MSR-VTT.**

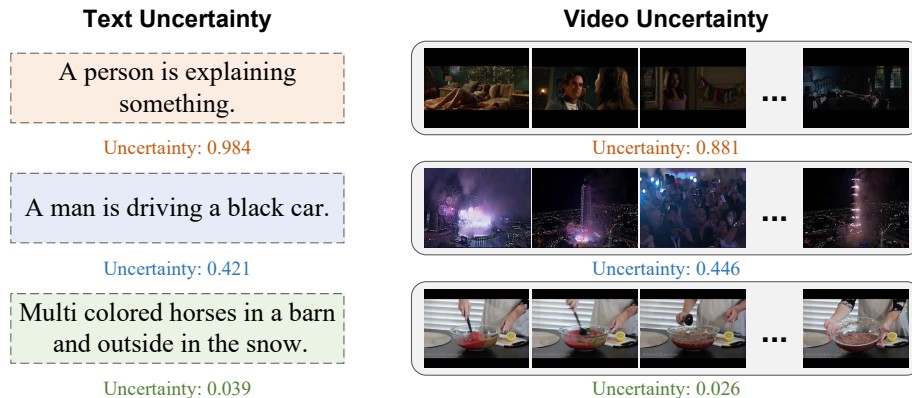

**Text Uncertainty**

A person is explaining something.
Uncertainty: 0.984

A man is driving a black car.
Uncertainty: 0.421

Multi colored horses in a barn and outside in the snow.
Uncertainty: 0.039

**Video Uncertainty**

Uncertainty: 0.881

Uncertainty: 0.446

Uncertainty: 0.026

Figure 6: **The visualization of data uncertainty on MSR-VTT.**

for text-to-video retrieval and a peak of 12.3 when $r = 700$ for video-to-text retrieval, respectively. The above phenomena prove PAU can indeed find uncertain data, which easily misleading the model into incorrect predictions, resulting in extremely unreliable retrieval results.

**Is the prediction more trustworthy?** To validate the effectiveness of PAU in providing trustworthy predictions, the performances of PAU and baseline are compared in the MSR-VTT's subsets under a series of different uncertain degrees. Precisely, $r$ samples with the highest uncertainty are selected to form the uncertain test subsets. As shown in Figure 5, PAU consistently outperforms the baseline with the largest gap of 8.6 (5.8) while $r = 70$ ($r = 120$) for t2v (v2t). The smaller the $r$ is, the higher the total uncertainty of the subset is, so the results confirm the ability of PAU to decrease the performance loss caused by uncertain data. Furthermore, PAU can provide trustworthy predictions even in high uncertainty sets.

**Visualization of data uncertainty.** Three captions and three videos with their uncertainty scores are shown in Figure 6. Obviously, the caption with high uncertainty is more semantically ambiguous, such as "A person is explaining something". By contrast, the caption with low uncertainty is more detailed, such as "Multi colored horses in a barn and outside in the snow". Moreover, the first video (complete video shown in Appendix. F) is more difficult to summarize due to the varied scenes, inducing high uncertainty. By contrast, the third video with low uncertainty is relatively simple.

## 5    Conclusion

In this work, we clearly define the inherent aleatoric uncertainty in multi-modal data. A Prototype-based Aleatoric Uncertainty Quantification (PAU) framework is designed to excavate the uncertain data and further provide trustworthy predictions. Uncertainty loss and diversity loss are proposed to support the uncertainty training by encouraging the learnable prototypes to represent the semantic subspace of varied modalities. Plentiful empirical experiments on a range of tasks and public benchmarks demonstrate the effectiveness and generalization of PAU.

**Broader impacts statement.** Cross-modal matching has become a common paradigm in large—scale multi-modal pretraining, such as CLIP [54]. However, the enormous data demand brings a huge challenge to pre-trained models. If we can accurately select high-quality data, which typically make greater contributions to the performance of pre-trained models, the pretraining process would become much more efficient. This work could be one of the first work to be aware of the aleatoric uncertainty in cross-modal retrieval. By quantifying uncertainty, higher quality data and more reliable results are provided, which could perhaps be extended to other multi-modal tasks in the future [2, 44, 63].

### Acknowledgements

This study is supported by grants from National Key R&D Program of China (2022YFC2009903/2022YFC2009900), the National Natural Science Foundation of China (Grant No. 62122018, No. 62020106008, No. 61772116, No. 61872064), Fok Ying-Tong Education Foundation(171106), and SongShan Laboratory YYJC012022019.

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

# Appendix

## A The Model Structure of the Baselines

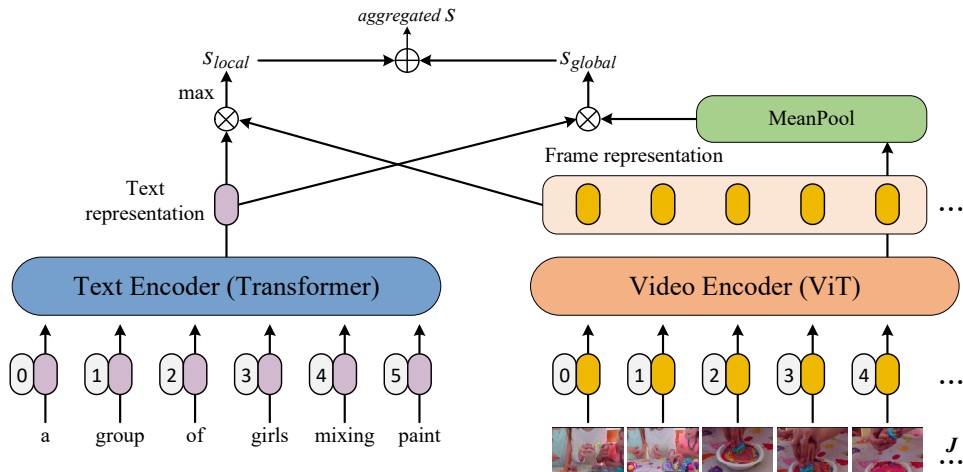

Figure 7: **The model structure of baseline on video-text retrieval.** The model takes a video-text pair as input. For the input video, we encode it into $J$ frame representations. For the input text, we encode it into a text representation. Then, a mean pooling layer aggregates $J$ frame representations to a global representation, which calculates the global similarity $s_{global}$ with the text representation. Meanwhile, the maximum value of the similarities between $J$ frame representations and the text representation is selected as $s_{local}$. Finally, we take the average of $s_{global}$ and $s_{local}$ as the final similarity $s$, which will be constrained by a cross-entropy loss in the training process. $\otimes$ means cosine similarity. $\oplus$ means average addition.

For image-text retrieval, we utilize CLIP [54] as the baseline straightly. As for video-text retrieval, the model structure is shown in Figure 7. To be specific, we follow the model structure of the CLIP4CLIP [49] to generate $J$ frame features $\mathbf{Z} = \{z_i^f | i = 1, 2, .., J\}$ for each video and a text feature $z^t$ for each caption. Afterwards, two branches are designed to catch global and local information, respectively. In global branch, We first adopt a mean pooling to aggregate $J$ frame features to get an 'average global feature' $\bar{z}^v$. Next, the cosine similarity $s_{global}$ between $\bar{z}^v$ and $z^t$ is calculated. In local branch, we first calculate a set of cosine similarities $S \in \mathbb{R}^J$ between $z^t$ and each frame feature $z_i^f$. Then we take the maximum value in $S$ as $s_{local}$. Finally, we take the average of $s_{global}$ and $s_{local}$ as the final similarity score. This similarity score will be constrained by a cross-entropy loss.

## B Evidence Generation

How to generate evidence $e$ from the similarity $s$ between the instance and the prototype is crucial for uncertainty modeling. Following [57], three optional functions, including ReLU function, softplus function, and exponential function, are used to generate evidence as follows:

**ReLU function**:

$$e(s) = \begin{cases} 0 & ,s \leq 0 \\ s & ,otherwise \end{cases} \tag{8}$$

**Softplus function**. Let $\theta$ denote a threshold of reverting to a linear function. $\gamma$ and $\theta$ are separately set to 1, 20 in the experiments. Softplus function can formulate as:

$$e(s) = \begin{cases} \dfrac{1}{\gamma} \log(1 + \exp(\gamma s)) & ,\gamma s \leq \theta \\ s & ,otherwise \end{cases} \tag{9}$$

**Exponential function**. Let $\tau$ be a scale parameter of the exponential function (Eq. 10), indicated as 5 in our experiments.

$$e(s) = \exp(s/\tau) \tag{10}$$

The performance comparison of three evidence-generation functions is shown in Table 7. It's worth noting that the exponential function outperforms other methods on multiple metrics. We therefore utilize it as the evidence-generation function in all experiments.

Table 7: Evidence generation function comparison on MSR-VTT

| Method | t2v | | | | | v2t | | | | |
|---|---|---|---|---|---|---|---|---|---|---|
| | R@1 | R@5 | R@10 | MdR | MnR | R@1 | R@5 | R@10 | MdR | MnR |
| Baseline | 45.6 | **72.9** | 81.5 | 2.0 | 14.5 | 45.2 | 71.6 | 81.5 | 2.0 | 10.9 |
| **ReLU** | 46.6 | 71.9 | 82.9 | 2.0 | 14.4 | 47.0 | 72.9 | 82.4 | 2.0 | 10.1 |
| **Softplus** | 47.6 | 72.7 | **83.0** | 2.0 | 14.4 | 47.8 | 71.9 | 82.9 | 2.0 | 10.3 |
| **Exponential** | **48.5** | 72.7 | 82.5 | **2.0** | **14.0** | **48.3** | **73.0** | **83.2** | **2.0** | **9.7** |

# C  Additional Experimental Results

## C.1  The number of prototypes

Table 8: Ablation study of prototype number $K$ on MSR-VTT

| Method | t2v | | | v2t | | | Mean |
|---|---|---|---|---|---|---|---|
| | R@1 | R@5 | R@10 | R@1 | R@5 | R@10 | R@1 |
| w/o PAU | 45.6 | 72.9 | 81.5 | 45.2 | 71.6 | 81.5 | 45.4 |
| K=1 | 47.2 | 72.2 | 81.4 | 46.2 | 72.7 | 81.3 | 46.7 |
| K=4 | 47.6 | 72.8 | 82.0 | 46.7 | 71.6 | 82.5 | 47.2 |
| K=8 | 48.5 | 72.7 | 82.5 | **48.3** | 73.0 | **83.2** | **48.4** |
| K=12 | **48.6** | 72.4 | **82.4** | 46.9 | 72.6 | 82.2 | 47.8 |
| K=16 | 47.8 | **73.0** | 82.0 | 47.0 | **72.8** | 82.1 | 47.4 |

Multiple prototypes are built to represent the overall semantics of the subspace. To figure out the best number of prototypes for representing subspace, the performance variety against prototype number $K$ is shown in Table 8. It can be seen that PAU always performs better than the baseline. Furthermore, R@1 of both t2v and v2t first increases to the peak against growing $K$ ($K = 12$ for t2v, $K = 8$ for v2t) and then decreases. To model the uncertainty consistently, $K$ is set to 8 for all modalities due to the maximum mean R@1 when $K = 8$.

## C.2  Different Initialization Methods for Prototypes

Table 9: Comparison of different prototype initialization methods on MSR-VTT

| Method | t2v | | | v2t | | |
|---|---|---|---|---|---|---|
| | R@1 | R@5 | R@10 | R@1 | R@5 | R@10 |
| kaiming_uniform [29] | 47.2 | 72.8 | 81.9 | 46.3 | 73.2 | 81.6 |
| kaiming_normal [29] | 46.8 | 72.5 | 82.2 | 46.6 | 73.0 | 82.3 |
| orthogonal [56] | 47.4 | **73.0** | **83.1** | 47.1 | **74.0** | **83.5** |
| Xavier [24] | **48.5** | 72.7 | 82.5 | **48.3** | 73.0 | 83.2 |

To explore the impact of different initial methods to prototypes, we employ several initialization methods for prototypes on MSR-VTT. The methods contain Xavier [24], kaiming_normal [29], kaiming_uniform [29], orthogonal [56] initialization. The results are shown below. We can observe that Xavier init methods can get the best R@1 performance on both t2v and v2t.

## C.3  Comparing Model Size and Complexity

The FLOPs and parameter number of PAU on MSR-VTT are computed to explore the computational costs and model complexity. We compare PAU with some previous methods, such as VATT [1],

Table 10: Comparison of model size and complexity on MSR-VTT.

| Method | Params (M) | FLOPs (G) |
|---|---|---|
| VATT [1] | 327.0 | 792.0 |
| Frozen [4] | 180.9 | 771.0 |
| MILES [23] | 180.9 | 771.0 |
| PAU | **84.2** | **36.0** |

Frozen [4], MILES [23]. The results are shown as below. It can be seen that the FLOPs and the parameter number of PAU are both significantly smaller than other methods, indicating that PAU is a light-weight approach and easy to deploy.

## C.4 Training with Noisy Correspondences

Table 11: Comparison on noisy MS-COCO[2]

| Noise ratio | Method | MS-COCO 1K | | | | | | MS-COCO 5K | | | | | |
|---|---|---|---|---|---|---|---|---|---|---|---|---|---|
| | | i2t | | | t2i | | | i2t | | | t2i | | |
| | | R@1 | R@5 | R@10 | R@1 | R@5 | R@10 | R@1 | R@5 | R@10 | R@1 | R@5 | R@10 |
| 0% | VSE∞ [8] | **82.0** | **97.2** | 98.9 | 69.0 | 92.6 | 96.8 | 62.3 | **87.1** | 93.3 | **48.2** | **76.7** | 85.5 |
| | PCME [11] | 80.1 | 96.6 | 98.7 | 67.6 | 92.1 | 96.9 | 59.9 | 85.8 | 92.3 | 46.1 | 75.0 | 84.6 |
| | PCME++ [10] | 81.6 | 97.2 | **99.0** | 69.2 | 92.8 | 97.1 | 62.1 | 86.8 | 93.3 | 48.1 | 76.7 | 85.5 |
| | **Baseline** | 80.1 | 95.7 | 98.2 | 67.1 | 91.4 | 96.6 | 62.9 | 84.9 | 91.6 | 46.5 | 73.8 | 82.9 |
| | **ours** | 80.4 | 96.2 | 98.5 | 67.7 | 91.8 | 96.6 | 63.6 | 85.2 | 92.2 | 46.8 | 74.4 | 83.7 |
| 20% | VSE∞ [8] | **78.4** | 94.3 | 97.0 | **65.5** | 89.3 | 94.1 | 58.6 | 83.4 | 89.9 | **45.0** | **72.9** | 81.7 |
| | PCME [11] | 76.8 | 95.4 | 98.2 | 63.8 | 90.5 | 96.0 | 55.7 | 81.7 | 90.0 | 41.8 | 71.4 | 81.7 |
| | PCME++ [10] | 78.4 | **95.9** | 98.4 | 64.9 | 90.8 | 96.1 | 57.7 | 83.9 | 91.0 | 43.2 | 72.3 | 82.4 |
| | **Baseline** | 76.0 | 94.3 | 97.5 | 63.4 | 89.0 | 94.8 | 55.3 | 79.1 | 90.4 | 41.0 | 68.8 | 79.3 |
| | **ours** | 78.2 | 95.2 | 98.1 | 64.5 | 90.0 | 95.4 | 59.3 | 82.9 | 90.4 | 44.2 | 71.3 | 81.3 |
| 50% | VSE∞ [8] | 44.3 | 76.1 | 86.9 | 34.0 | 69.2 | 84.5 | 22.4 | 48.2 | 61.1 | 15.8 | 38.8 | 52.1 |
| | PCME [11] | 73.4 | 94.0 | 97.6 | 59.8 | 88.3 | 94.8 | 50.9 | 78.8 | 87.3 | 37.9 | 67.3 | 78.4 |
| | PCME++ [10] | 74.8 | **94.3** | 97.7 | 60.4 | **88.7** | 95.0 | 52.5 | 79.6 | 88.4 | 38.6 | 68.0 | 79.0 |
| | **Baseline** | 73.9 | 93.0 | 97.2 | 60.1 | 87.3 | 94.0 | 54.1 | 78.5 | 86.6 | 39.7 | 67.2 | 77.5 |
| | **ours** | 76.4 | 94.1 | 97.6 | 62.3 | 88.5 | 94.6 | 57.3 | 81.5 | 88.8 | 41.9 | 69.4 | 79.6 |

We additionally compare PAU with some CLIP ViT-B/32 backbone-based methods [8, 11, 10, 54] fine-tuned on noisy MS-COCO dataset. The results of training on 0%, 20%, and 50% noise ratios are reported in the Table 11. We can observe that our method exhibits significant advantages while training with several noisy correspondences, especially on 50% noise ratio. Although the uncertainty-aware method, PAU, is not designed for tackling the noisy correspondences, it still successfully handles the scenario, indicating that PAU possesses powerful robust learning ability even while training with noisy correspondences.

## D   The Explanation of High Boost on MSVD

In Table 2, we find the performance improvement of MSVD on v2t is significantly larger than other benchmarks. Investigating its reason, the MSVD is a tiny dataset with only 1,200 videos for the training set and approximately 40 captions per video (48,000 captions in total). In this case, utilizing cross-entropy loss, which views the diagonal pairs as positives, will lead into tremendous noise, especially with a large batchsize. In a batch with the size of 256, the probability of any texts coming from different videos is about $3.49 \times 10^{-13}$. The solution is as follows.

$$P = \frac{\prod_{i=1}^{B} C_{m_i}^1}{C_N^B}. \tag{11}$$

The probability of any texts coming from different videos can denote as Eq. 11, where $B$ is the batchsize of 256, and $N$ is the total caption number of 48,000. $m_i$ indicates the candidate number

---

[2]Results come from: https://github.com/naver-ai/pcmepp/issues/2

of $i^{th}$ text in a batch with $m_i = 48000 - 40 \times (i - 1)$ in MSVD. Then the probability $P$ can be calculated by:

$$P = \frac{\prod_{i=1}^{256} C_{48000-40\times(i-1)}^{1}}{C_{48000}^{256}}$$

$$\log P = \sum_{i=1}^{256} \log(48000 - 40 \times (i - 1)) - \sum_{i=1}^{256} \log(48000 - i)$$

$$\log P = (\log 48000 + \log 47960 + \log 47920 + \cdots + \log 37800)$$
$$- (\log 48000 + \log 47999 + \log 47998 + \cdots + \log 47745)$$

$$\log P \approx -28.685$$

$$P \approx -3.4910^{-13}.$$

(12)

The resultant noises mislead the model training, resulting in a terrible performance reduction. Our uncertainty-aware approach can mitigate the detriment caused by these data and improve the model robustness effectively.

## E  Proof of Ambiguous Data with High Information Entropy

In Section. 1, we propose a theorem that "the ambiguous multi-modal data highly associated with multiple semantics, such as fast-paced videos and non-detailed texts, lead to high information entropy, standing for high uncertainty." The mathematical *Theorem* and *Proof* are given below.

**Theorem.** Suppose $z \in \mathbb{R}^D$ is the vector of multi-modal data. Assuming the modality semantic subspace can be divided into $K$ finite categories as $X = \{x_i \in \mathbb{R}^D : i = 1, 2, \cdots, K\}$. The information entropy is denoted as Eq. 13:

$$\mathcal{H}(X) = -\sum_{i=1}^{K} p(x_i) \log p(x_i) \quad \text{s.t.} \quad \sum_{i=1}^{K} p(x_i) = 1,$$

(13)

where $p(x_i)$ is the $i^{th}$ semantics probability derived from cosine similarity $cos(z, x_i)$ through a softmax layer. Let $Q \sim q(x)$ be a Discrete Uniform Distribution [14], then we have:

$$\mathcal{J}(P \parallel Q) \propto \frac{1}{\mathcal{H}(X)},$$

(14)

where $\mathcal{J}(P \parallel Q)$ denotes the Jensen–Shannon Divergence (JSD) [16], a metric to measure the distance between two probability distributions. Namely, Eq. 14 means that the closer the distribution $P \sim p(x)$ is to the Uniform Distribution $Q \sim q(x)$, the higher the information entropy $\mathcal{H}(X)$ is.

**Proof.** Let's first figure out what distribution of information entropy is the largest. Given an objective function (Eq. 15) and its constraint (Eq. 13):

$$\max_{P} \mathcal{H}(X) = -p(x_1) \log p(x_1) - p(x_2) \log p(x_2) - ... - p(x_K) \log p(x_K).$$

(15)

Let $\lambda$ denote Lagrange multiplier. The Lagrangian function can be constructed as:

$$\ell(p, \lambda) = -\sum_{i=1}^{K} p(x_i) \log p(x_i) + \lambda(\sum_{i=1}^{K} p(x_i) - 1).$$

(16)

Separately computing the partial derivatives for $p(x_1), p(x_2), ..., p(x_K)$:

$$\frac{\partial \ell(p, \lambda)}{\partial p(x_i)} = \lambda - \log p(x_i) = 0.$$

(17)

From this:

$$\log p(x_1) = \log p(x_2) = ... = \log p(x_K) = \lambda.$$

(18)

Considering the constraints of Eq. 13, the $p(x)$ essentially satisfy:

$$p(x_1) = p(x_2) = ... = p(x_K) = \frac{1}{K},$$

(19)

which is the density of discrete uniform distribution. Therefore, the $\mathcal{H}(X)$ gets the great value $\log K$ when the probability distribution conforms to a Discrete Uniform Distribution [14].

Then Jensen–Shannon Divergence (JSD) [16] is used to measure the distance between the probability distribution $P \sim p(x)$ and the Discrete Uniform Distribution $Q \sim q(x)$.

$$
\begin{aligned}
\mathcal{J}(P \parallel Q) &= \frac{1}{2} \sum_{i=1}^{K} p(x_i) \log(\frac{p(x_i)}{p(x_i) + q(x_i)}) + \frac{1}{2} \sum_{i=1}^{K} q(x_i) \log(\frac{q(x_i)}{p(x_i) + q(x_i)}) + 1 \\
&= \frac{1}{2}(\sum_{i=1}^{K} p(x_i) \log p(x_i) - \sum_{i=1}^{K} p(x_i) \log(p(x_i) + q(x_i)) \\
&\quad + \sum_{i=1}^{K} q(x_i) \log q(x_i) - \sum_{i=1}^{K} q(x_i) \log(p(x_i) + q(x_i))) + 1.
\end{aligned}
\tag{20}
$$

Since $q(x)$ conforms to the Discrete Uniform Distribution, $q(x_i) = \frac{1}{K}$ and $\sum_{i=1}^{K} q(x_i) \log q(x_i) = -\log K$. The JSD function (Eq. 20) can be converted to:

$$
\begin{aligned}
\mathcal{J}(P \parallel Q) &= \frac{1}{2}(\sum_{i=1}^{K} p(x_i) \log p(x_i) - \sum_{i=1}^{K} p(x_i) \log(\frac{1}{K} + p(x_i)) \\
&\quad - \log K - \frac{1}{K} \sum_{i=1}^{K} \log(\frac{1}{K} + p(x_i))) + 1 \\
&= \frac{1}{2}(\sum_{i=1}^{K} p(x_i) \log p(x_i) - \sum_{i=1}^{K} (\frac{1}{K} + p(x_i)) \log(\frac{1}{K} + p(x_i)) - \log K) + 1.
\end{aligned}
\tag{21}
$$

Note that there are two components affecting the $\mathcal{J}(P \parallel Q)$, $\sum_{i=1}^{K} p(x_i) \log p(x_i)$ and $\sum_{i=1}^{K} (\frac{1}{K} + p(x_i)) \log(\frac{1}{K} + p(x_i))$. For convenience, let:

$$
\begin{aligned}
A(P) &= -\mathcal{H}_{P \sim p(x)}(X) = \sum_{i=1}^{K} p(x_i) \log p(x_i), \\
B(P) &= \sum_{i=1}^{K} (\frac{1}{K} + p(x_i)) \log(\frac{1}{K} + p(x_i)).
\end{aligned}
\tag{22}
$$

It can be seen that $\mathcal{J}(P \parallel Q) \propto (A(P) - B(P))$. When $P \sim p(x)$ is closer to the Discrete Uniform Distribution $Q \sim q(x)$, i.e., $\mathcal{J}(P \parallel Q) \downarrow$, three cases are possible for $A(P)$ and $B(P)$:

*a.* $A(P) \downarrow$ *and* $B(P) \downarrow$, *but the former has a larger reduction, i.e.,* $\Delta A(P) - \Delta B(P) < 0$.

*b.* $A(P) \downarrow$ *and* $B(P) \uparrow$.

*c.* $A(P) \uparrow$ *and* $B(P) \uparrow$, *but the latter has a larger growth, i.e.,* $\Delta A(P) - \Delta B(P) < 0$.

$\Delta \to 0$ is the amount of function change. $A(P)$ are both declining in cases of *a* and *b*, reflecting the increase of $H_{Q \sim q(x)}(X)$ according to Eq. 22. Thus, only case *c* possibly induces the decrease of $H_{Q \sim q(x)}(X)$ with the decrease of $\mathcal{J}(P \parallel Q)$. If there is no case suit *c*, the theorem is proved.

Let $f(x) = x \log x$. Its first-order derivative and second-order derivative are $f(x)' = \log x + 1$ and $f(x)'' = \frac{1}{x} > 0$, respectively. It is a convex function obtaining the minimal value at $x = \frac{1}{e}$.

Due to $\sum_{i=1}^{K} p(x_i) = 1$, when $p(x_a) \to p(x_a) + \Delta$, there must be a $p(x_b) \to p(x_b) - \Delta$. Subject to $A(P) \uparrow$, Eq. 23 should be satisfied:

$$
\Delta A(P) = f(p(x_a))' - f(p(x_b))' = \log \frac{p(x_a)}{p(x_b)} > 0 \implies p(x_a) > p(x_b).
\tag{23}
$$

When $0 < p(x_b) - \Delta < p(x_b) < p(x_a) < p(x_a) + \Delta < 1$, we have:

$$
\begin{aligned}
\Delta A(P) - \Delta B(P) &= (f(p(x_a))' - f(p(x_b))') - (f(p(x_a) + \frac{1}{K})' - f(p(x_b) + \frac{1}{K})') \\
&= \log \frac{p(x_a)}{p(x_a) + \frac{1}{K}} - \log \frac{p(x_b)}{p(x_b) + \frac{1}{K}} \\
&= \log(1 + \frac{1}{Kp(x_b)}) - \log(1 + \frac{1}{Kp(x_a)}) > 0.
\end{aligned}
\tag{24}
$$

It indicates that case $c$ is an impossible case when $\mathcal{J}(P \parallel Q) \downarrow$. Thus, only $a$ and $b$ are possible when $\mathcal{J}(P \parallel Q) \downarrow$. In these two cases,

$$
\mathcal{J}(P \parallel Q) \downarrow \longrightarrow A(P) \downarrow \longrightarrow H_{P \sim p(x)}(X) \uparrow.
\tag{25}
$$

Therefore,

$$
\mathcal{J}(P \parallel Q) \propto \frac{1}{\mathcal{H}(X)},
\tag{26}
$$

demonstrating that the closer to a uniform distribution, the larger the information entropy is. It concludes the proof.

# F    The Detailed Video Visualization

In Figure 6 of the main paper, we present several video fragments with their uncertainty. In Figure 8, more detailed videos are displayed to illustrate the uncertainty degree of the videos.

Figure 8(a) is a video with a 0.881 uncertainty score. Actually, it is difficult to understand the meaning expressed in the video. Obviously, as the uncertainty diminishes, the number of scenes in the videos decreases, and the videos become easier to understand.

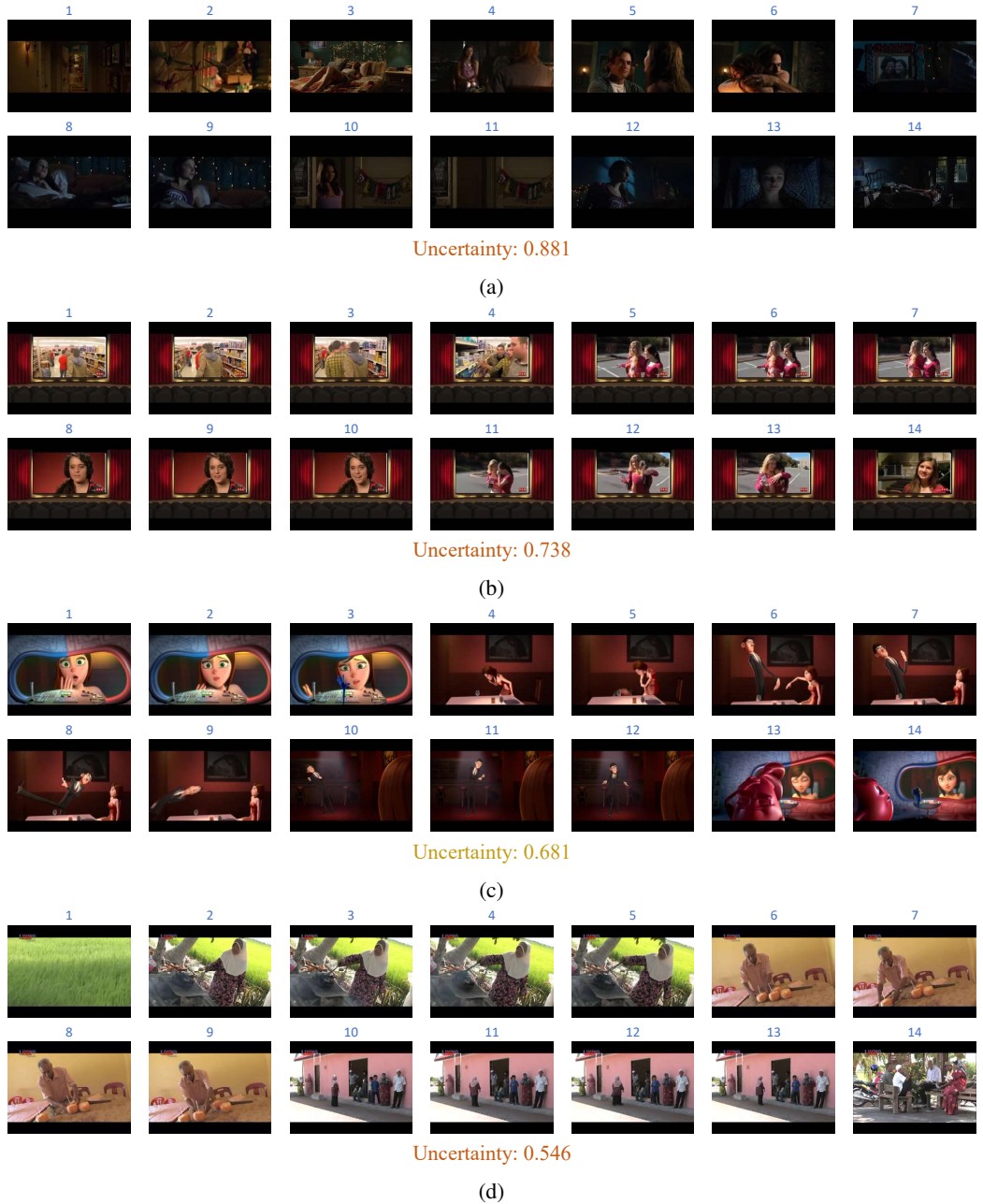

Uncertainty: 0.881

(a)

Uncertainty: 0.738

(b)

Uncertainty: 0.681

(c)

Uncertainty: 0.546

(d)

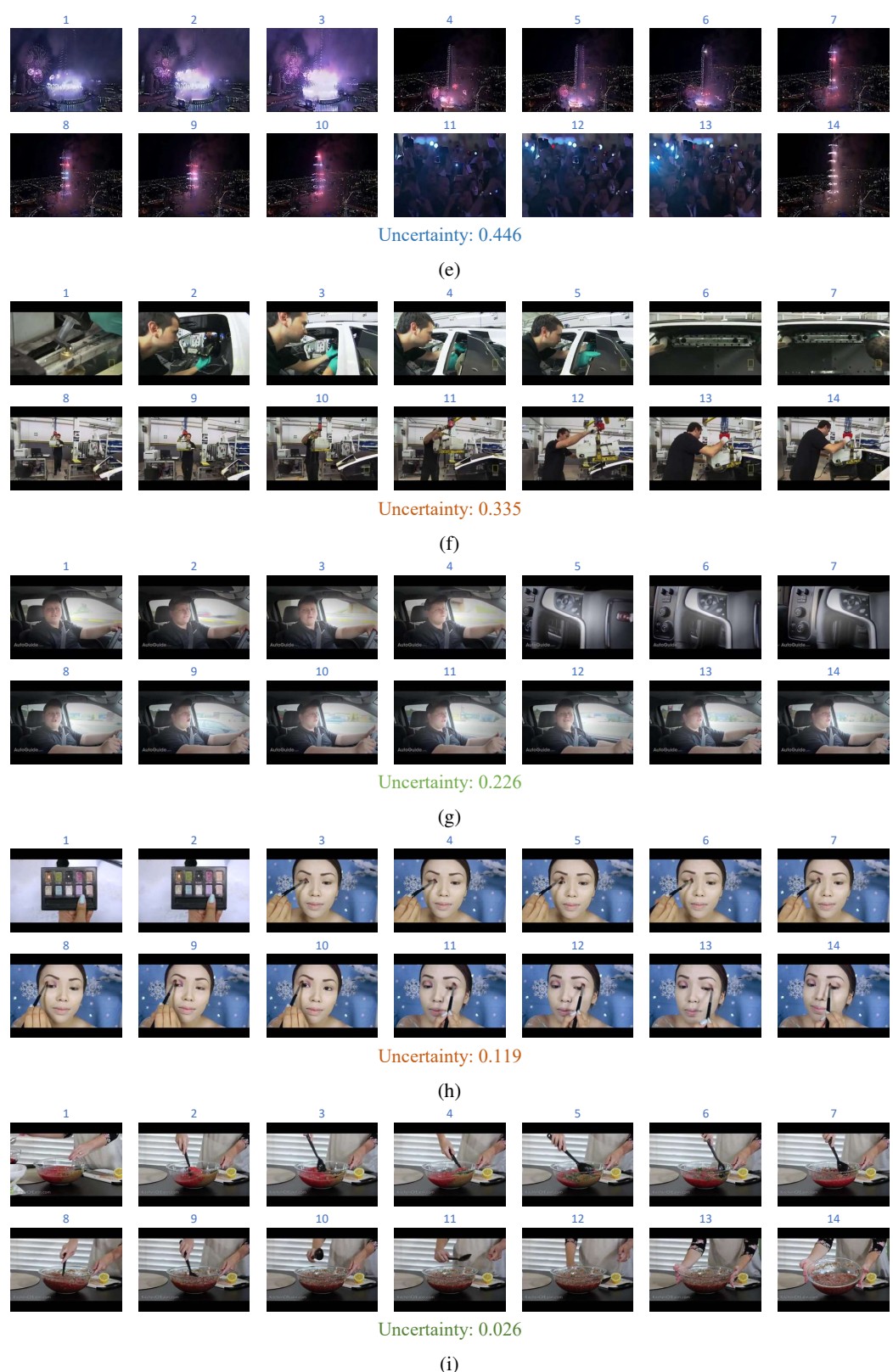

Figure 8: The detailed video visualization of data uncertainty on MSR-VTT.

