# OpenReview forum: "Prototype-based Aleatoric Uncertainty Quantification for Cross-modal Retrieval"
_NeurIPS.cc/2023/Conference — NeurIPS 2023 poster_

### Official Review · Reviewer_hqQV · 2023-07-03

**Soundness:** 3 good
**Presentation:** 3 good
**Contribution:** 3 good
**Rating:** 6
**Confidence:** 3

**Summary:**

The paper focuses on the unreliable prediction of cross-modal retrieval methods caused by low-quality data. The authors introduce evidential theoretical framework to quantify the uncertainty arising from inherent data ambiguity.  To provide trustworthy uncertainty predictions, they propose a novel Prototype-based Aleatoric Uncertainty Quantification (PAU) framework. Extensive experiments demonstrate the effectiveness of their methods.

**Strengths:**

- To the best of my knowledge, this paper is the first to combine the evidential framework with cross-modal retrieval methods, so this work is quite novel.
- The idea of introducing uncertainty quantification to mitigate the negative impact of low-quality data on retrieval is interesting and sound. The experimental evaluations in the paper also provide strong evidence to support the claims made in the paper. These experiments are wide-ranging and cover several different datasets.
- The paper is well written and easy to follow. However, there is room for improvements as I mention below.

**Weaknesses:**

1.  While the experiments are quite wide-ranging, the details provided in the paper are quite scarce, such as how many epochs are trained and how to choose hyperparameters. Code is also not provided.

2.  Although the authors provide related work on cross-modal retrieval and uncertainty quantification, it is not clear how these works differ from their proposed method.

3. There are some unclear points in the paper.
- What is the meaning of the histogram in Figure 1(b)?
- Does PAU only contain the component of $L_{uct}$, $L_{div}$ and Re-rank? How does it constrain whether $M''$ is predicted accurately? Providing pseudocode would help readers better understand the proposed method.

**Questions:**

1. What is the effect of using different initialization methods for all prototypes?
2. What is the time complexity of the algorithm? How does it compare to other algorithms?
3. The authors argue that the uncertainty loss forces the prototypes into learning the rich semantics of subspace to realize accurate uncertainty quantification. But I'm confused that the uncertainty loss only limits the consistency of U and H, why it can achieve accurate uncertainty quantification?

**Limitations:**

The authors point out the limitations of their work.

---

> ### Author Rebuttal · Authors · 2023-08-09
>
> We thank the reviewer hqQV for the positive, valuable and detailed review, as well as the constructive suggestions for improvement. Our responses to the reviewer’s questions are below:
>
> ***Q1 : Some details about the number of training epochs, how to choose parameters, and code are scarce.***
>
> **A1:** Thanks for the constructive suggestions. You can find the training epochs in Section 4.2 (Implementation details). PAU is trained with 5, 3, 20, and 5 epochs on MSR-VTT, MSVD, DiDeMo, and MS-COCO, respectively. The code can be found in the url link of the abstract. As for the parameters chosen, there are two important parameters for PAU, prototype number $K$ and re-ranking control parameter $\beta$. For parameter $K$, we perform experiments on three datasets with different $K$ numbers. The results are shown below. We can observe that PAU obtains the best Mean R@1 performance when $K=8$ on all benchmarks. Thus, we select $K=8$. For control parameters $\beta$, it is a learnable parameter. After the model is trained, we freeze other parameters of the model and only train $\beta$ using the re-ranked similarity matrix constrained by a cross-entropy loss for few steps.
> |||MSR|VTT||||||DiDeMo||||||MSVD|||
> |-|-|-|-|-|-|-|-|-|-|-|-|-|-|-|-|-|-|
> |||**t2v**|**v2t**||**Mean**|||**t2v**|**v2t**||**Mean**|||**t2v**|**v2t**||**Mean**|
> ||**R1**|**R5**|**R1**|**R5**|**R1**||**R1**|**R5**|**R1**|**R5**|**R1**||**R1**|**R5**| **R1**|**R5**|**R1**|
> |K=1|47.2|72.2|46.1|72.7|46.7||46.9|75.9|46.1|74.6|46.5||46.4|76.9|64.8|93.2|55.6|
> |K=4|47.6|72.8|46.7|71.6|47.2||48.0|75.8|47.3|74.8|47.65||46.6|77.0|67.9|92.9|57.25|
> |K=8|48.5|72.7|**48.3**|**73.0**|**48.4**||**48.6**|76.0|**48.1**|74.2|**48.35**|| **47.3**|**77.4**|**68.9**|**93.1**|**58.1**|
> |K=12|**48.6**|72.4|46.9|72.6|47.8||48.1|75.8|46.2|75.4|47.15||46.9|77.1|68.3|92.2| 57.6   |
> |K=16|47.8|**73.0**|47.0|72.8|47.4||**48.6**|**76.9**|47.7|**75.5**|48.15||47.0|**77.4**|**68.9**|92.5|57.95|
>
> ***Q2 : How do these related works differ from PAU?***
>
> **A2:** Previous cross-modal retrieval methods only focus on predicting the relevance of vision-language pairs. However, these methods don't consider the reliability of predictions, which will be decreased by low-quality input data. Previous uncertainty quantification methods mainly focus on single-modal tasks, such as classification and segmentation tasks. The gap between single-modal tasks and multi-modal tasks makes previous uncertainty quantification methods hardly transfer to multi-modal tasks.
>
> ***Q3 : What is the meaning of the histogram in Figure 1(b)?***
>
> **A3:** We are sorry for the lack of legends in Figure 1(b). The top legend should be Text A, and the bottom legend should be Text B. The histogram means the non-detailed text B has high similarities with more semantics categories than detailed text A. You can find the correctly updated **Figure 7 attached in pdf file of common response**.
>
> ***Q4 : How do PAU components constrain $M''$ to be predicted accurately?***
>
> **A4:** PAU only contains the components of $L_{uct}$, $L_{div}$, and re-rank. In the re-rank phase, there are two learnable parameters $\beta_1$ and $\beta_2$, which will constrain $M''$ predicted accurately. Concretely, after the model is trained, we freeze other parameters of the model and only train $\beta$ using the re-ranked similarity matrix with the constraint by cross-entropy loss. The cross-entropy loss will make the $M''$ accurate by learning proper parameters $\beta$. Limited by the rebuttal character number, we will add the pseudo code of $M''$ computing in our final paper.
>
> ***Q5 : What is the effect of using different initialization methods for all prototypes?***
>
> **A5:** We test PAU with several initialization methods for prototypes on MSR-VTT. The methods contain Xavier [9], kaiming_normal [7], kaiming_uniform [7], and orthogonal [8] initialization. The results are shown in the Table below. We can observe that Xavier init method can get the best R@1 performance on both t2v and v2t.
> |||**t2v**|||**v2t**||
> |-|-|-|-|-|-|-|
> |**Method**|**R1**|**R5**|**R10**|**R1**|**R5**|**R10**|
> |kaiming_uniform|47.2|72.8|81.9|46.3|73.2|81.6|
> |kaiming_normal|46.8|72.5|82.2|46.6|73.0|82.3|
> |orthogonal|47.4|**73.0**|**83.1**|47.1|**74.0**|**83.5**|
> |Xavier|**48.5**|72.7| 82.5|**48.3**|73.0|83.2|
>
> ***Q6 : What is the time complexity of the algorithm? How does it compare to other algorithms?***
>
> **A6:** We compute the FLOPs and parameter number of PAU on MSR-VTT. Meanwhile, we compare our methods with some famous previous methods. The results are shown in the Table below. We can observe that the FLOPs and the parameter number of PAU are both significantly smaller than other methods, which means that PAU is a light-weight approach and easy to deploy.
> |Methods|FLOPs (G)|Params (M) |
> |-|-|-|
> |VATT [10]|792|327|
> |Frozen [11]|771|180.9|
> |MILES [12]|771| 180.9|
> |PAU|**36**|**84.2**|
>
> ***Q7 : The uncertainty loss only limits the consistency of U and H, why it can achieve accurate uncertainty quantification?***
>
> **A7:** In this paper, we define that an uncertain instance should have higher mean similarities with the instances from the other modality. For example, given a specific image $i$ with the uncertainty score $u_i$. $h_i$ denotes the mean similarity between $i$ and all texts. According to the definition before, **$u_i$ and $h_i$ should be positively correlated**. In the actual training process, it is hard to compute the mean similarity between an image and all texts in the datasets because of the high computational cost. Thus, we only compute the mean similarity between an image and the texts in a batch. Assuming $batchsize=N$, all images' uncertainty scores make up $U=[u_1, u_2, ..., u_N]$, and the mean similarity between these images and all texts make up $H=[h_1, h_2, ..., h_N]$. By limiting the consistency of $U$ and $H$, we can force the $u$ and $h$ to be positively correlated. Then we can obtain the precise uncertainty estimator after training.

---

> > ### Comment · Reviewer_hqQV · 2023-08-16
> >
> > Thank you for your rebuttal. After reading all rebuttals, I maintain my score.

---

> > > ### Author Response · Authors · 2023-08-17
> > >
> > > Thanks to the Reviewer hqQV for the positive response. All the additional experiments and discussion will be supplemented in our revised paper. If you have any other concerns, we are happy to address them.

---

### Official Review · Reviewer_b4gq · 2023-07-05

**Soundness:** 3 good
**Presentation:** 3 good
**Contribution:** 4 excellent
**Rating:** 8
**Confidence:** 5

**Summary:**

This paper addresses the aleatoric uncertainty quantification in cross-modal retrieval. The authors point out that the model predictions are often unreliable due to the aleatoric uncertainty, which is induced by low-quality data. They first define the inherent aleatoric uncertainty in multi-modal data. Then K prototypes are constructed as the belief masses to employ DST and SL, two classical theories in uncertainty quantification. On top of this, the uncertainty loss and diversity loss are proposed to help prototypes represent the overall semantic space. Extensive experiments are conducted on multiple tasks and benchmarks.

**Strengths:**

1.	This work presents a novel insight for the uncertainty of cross-modal retrieval. They classify uncertainty into aleatoric uncertainty and epistemic uncertainty, and give a clear definition of aleatoric uncertainty for multi-modal data. I think it will inspire further uncertainty-aware work in cross-modal retrieval or multi-modal learning.
2.	In Appendix, the proof of “the ambiguous multi-modal data highly associated with multiple semantics” is intuitive and reasonable.
3.	The experiments greatly prove the claims, especially Fig. 3 and Fig. 4. These two experiments reflect that the uncertain data indeed have a significant negative impact on results, and PAU could provide more trustworthy predictions. This phenomenon may be able to give an inspiring insight for robust training.

**Weaknesses:**

1.	In the reranking process, beta_1 and beta_2 are two learnable parameters. But it is unclear how these parameters are trained in this paper.
2.	Prototypes are expected to represent the overall semantics space. Do they really represent the overall semantics space after training?
3.	The paper only mentions uct loss and div loss, which seem to be proxy-based loss functions, and can not optimize the prediction directly like cross-entropy loss or triplet loss. I wonder if other basic losses are used in the training process.
4.	The Table 7 in appendix shows that the exponential function is the best method to generate evidence. Why there is a large performance gap between ReLu function and exponential function? Or what is the advantages of exponential function.
5.	The legend in Figure 1 (b) is empty.

**Questions:**

Refer to weaknesses.

**Limitations:**

Yes

---

> ### Author Rebuttal · Authors · 2023-08-09
>
> We would like to thank the reviewer b4gq for providing professional and valuable feedback, as well as encouraging comments. In what follows, we wish to address some of the questions you raised as below.
>
> ***Q1 : How learnable parameters $\beta_1$ and $\beta_2$ are learned in PAU?***
>
> **A1:** Thanks for the valuable question. $\beta_1$ and $\beta_2$ are two learnable parameters to control the impact degree of re-ranking. After the model is trained, we first freeze other parameters of the model. Then, we utilize Eq. 7 to re-rank the similarity matrix. Finally, we re-trained the model for few steps constrained by a normal cross-entropy loss. After re-training, we can obtain the learned parameters $\beta_1$ and $\beta_2$, which can output the best re-ranked predictions.
>
> ***Q2 : Prototypes are expected to represent the overall semantics space. Do they really represent the overall semantics space after training?***
>
> **A2:** Thanks for the constructive suggestions. To prove that, we carry out an experiment of pearson correlation analysis. The pearson correlation coefficient (**PCC**) $r_{xy}$ ($r_{xy} \in [-1, 1]$) is a metric to estimate the strength of the correlation between variable $x$ and variable $y$. Generally, $r_{xy} > 0.5$ means that two variables are strongly positively correlated. Thus, given an instance, we compute the pearson correlation coefficient between its uncertainty score $u$ and its mean similarity $h$ with the instances from the other modality on two datasets. If $u$ and $h$ are strongly positively correlated, this means that the prototypes represent the overall semantics space well. The results are shown in the Table below.
>
> | Method | PCC-V  | PCC-T  | Dataset  |
> |:------:|:------:|:------:|:--------:|
> | PAU    | 0.886  | 0.786  | MSCOCO   |
> | PAU    | 0.917  | 0.939  | MSR-VTT  |
>
>  It can be seen that $u$ and $h$ are all strongly positively correlated on both the image-text retrieval task and video-text retrieval task, which means that the prototypes in PAU indeed represent the overall semantics space.
>
> ***Q3 : If other basic losses are used in the training process?***
>
> **A3:** Yes, uncertainty loss and diversity loss can be regarded as two proxy loss functions. We also utilize the cross-entropy loss to constrain the similarity matrix. The final loss formulation should be $\mathcal L_{total}=\mathcal L_{CE} + \mathcal L_{uct} + \mathcal L_{div}$.
>
> ***Q4 : Why does the exponential function outperform than softplus function and ReLU function?***
>
> **A4:** The experiment results in **Table. 7 of the Appendix** show that the performance using exponential function > softplus function > ReLU function. The reason of exponential function outperforms than other two functions is that after similarity $s$ > $\theta$, ReLU function and softplus function will convert to a linear function. For the linear function, the evidence change of $s$ boosted from 0.2 to 0.3 will be the same as the evidence change of $s$ boosted from 0.9 to 1. But the latter boost is harder than the former, and it should take a higher evidence boost. The reason of softplus function outperforms than ReLU function is that $e=0$ when $s < 0$ in ReLU function, which will discard half information from $s$.
>
> ***Q5 : The legend in Figure 1(b) is empty.***
>
> **A5:** Thanks for the detailed review. We are sorry for the lack of legends in Figure 1(b). The top legend should be Text A, and the bottom legend should be Text B. You can find the correctly updated **Figure 7 in the attached pdf file of common response**.

---

> > ### Comment · Reviewer_b4gq · 2023-08-16
> >
> > My concerns are well addressed. I insist on accepting this work.

---

> > > ### Author Response · Authors · 2023-08-17
> > >
> > > Thanks to the Reviewer b4gq for the positive response. All the additional experiments and discussion will be supplemented in our revised paper.

---

### Official Review · Reviewer_tGbU · 2023-07-06

**Soundness:** 2 fair
**Presentation:** 2 fair
**Contribution:** 2 fair
**Rating:** 3
**Confidence:** 4

**Summary:**

This paper focuses on improving video-text retrieval by considering the uncertainty (or confidence) when computing image-text similarity. The high-level idea is to estimate the data uncertainty and then perform re-ranking to improve the performance of retrieval.

The authors designed a PAU framework, Prototype-based Aleatoric Uncertainty Quantification with CLIP encoders, to estimate the uncertainty score of the input instances. The authors conducted task-specific fine-tuning on the target datasets, and reported marginal performance improvements.

**Strengths:**

- The authors addressed an interesting problem of estimating the uncertainty of data, which is used to improve retrieval performance.
- The proposed PAU framework is presented clearly.
- Both the uncertainty loss and diversity loss are technically valid and straightforward.
- The authors conducted several experiments on multiple benchmarks, and demonstrate marginal performance improvements.

**Weaknesses:**

- The goal and task definition of this paper seem not clear. Both the introduction section and Figure 1 are somewhat confusing, which makes me difficult to get the main point. An image (or a single video frame) should contain rich semantics. Typically, an image is paired with a caption (or sentence), forming a image-text pair, to describe the semantics. However, the authors seem to argue that a single-scene video should match only a single word (i.e., talking), which is counter-intuitive. The multi-scene video may have richer semantics than single-scene video, but the authors argue that matching multiple semantics is not ideal. Overall, their key idea is counter-intuitive.
- The authors conducted experiments solely on downstream fine-tuning. The fine-tuning datasets are usually cleaner and with higher quality. A majarity of the noisy image-text paired data should be found in VL pre-training dataset. While downstream fine-tuning data (like youcook2 or MSRVTT) are human annotated/collected, the pre-training data is web-crawled noisy imgae-text pairs. To study the capability of uncertainty estimation for low-quality data, it would be much more interesting to experiment their proposed method in the pre-training stage. The proposed method may have potential to greatly improve the pre-training, but it remains unknown in current stage.
- The paper is not written well, making it difficult to understand the motivation. This is probably due to unprecise or uncommon wording in the writing.

**Questions:**

see weakness

**Limitations:**

No potential negative societal impact

---

> ### Author Rebuttal · Authors · 2023-08-09
>
> We thank the reviewer tGbU for providing constructive feedback. In what follows, we wish to address the questions you raised as below.
>
> ***Q1 : The authors seem to argue that a single-scene video should match only a single word (i.e., talking), which is counter-intuitive.***
>
> **A1:** We are sorry for causing some confusion to you. Actually, we don't argue that a single-scene video should match only a single word (i.e., talking). The example of **Video A in Figure 1(a)** matching the word **"talking"** is just one instance. It can also match the caption **"two people are talking"**. However, whether it is "talking" or "two people are talking", they both similarly convey the summary semantics of **"talking"**. But the multi-scene videos are hard to be summarized to a single semantics because the frames describing different scenes can match different semantics. The focus of this example is only to **illustrate the difference** in semantic richness between single-scene videos and multi-scene videos.
>
> ***Q2 : The multi-scene video may have richer semantics than single-scene video, but the authors argue that matching multiple semantics is not ideal.***
>
> **A2:** Thanks for your comment. We don't argue that the multi-scene video matching multiple semantics is not ideal. By contrast, we claim that multi-scene videos are easy to match with multiple semantics. For example, in **Figure. 1(a) of the paper**, single-scene video A can be briefly summarized as the simple word **"talking"** or the brief caption **"two people are talking"**. In contrast, the multi-scene video B is hard to summarize briefly because of its rich semantics, leading to confusion in the retrieval process, i.e., it is hard to be described using a simple description. Thus, this confusion is the **uncertainty** we defined in our paper. The primary concept of this work is to identify these easily confused instances and mitigate their impact on the retrieval process. We use uncertain scores to **quantify the confusion degree these instances introduce** during the retrieval process. It is worth noting that our work is to quantify the uncertainty of individual instances (a video, an image,  or a text), not a vision-language pair. But we can compute the confidence $c$ of a vision-language pair using $c=(1-u_v)\times(1-u_t)$. $u_v$ denotes the uncertainty score of the video in this pair. $u_t$ denotes the uncertainty score of the text in this pair.
>
> ***Q3 : Training in the pre-training phase on web-crawled noisy dataset.***
>
> **A3:** Thanks for the constructive suggestions. However, limited by the high computation cost and time, it is hard to test our method in the pre-training phase. We will achieve it in the future work. Instead, we train our method on the noisy dataset and compare it with other **CLIP ViT-B/32 backbone-based methods**. We follow the noise split of NCR [4] with **20%, 50%** relationships randomly shuffled. It can be seen that PAU outperforms all other methods with a significant gap in 20% and 50% noise ratios. This clearly demonstrates that PAU exhibits robust learning capabilities as well.
>
> **Note:** CLIP is our baseline.
>
> |           |          | AVG R1    |            |
> |-----------|----------|-----------|------------|
> | **Method**| **0% noise**|**20% noise**| **50% noise**  |
> | VSE$\infty$ [5]| **55.2**| 51.4      | 18.4       |
> | PCME [6]     | 53.0     | 48.1      | 43.0       |
> | NCR [4]      | -        | 48.8      | 45.5       |
> | CLIP [1]     | 54.7     | 48.3      | 46.8       |
> | PAU       | **55.2** | **51.8**  |**49.6**    |
>
> ***Q4 : Some unprecise and uncommon wordings make motivation be understood hard.***
>
> **A4:** Thanks for your comment. We guess that some wording like **"fickle video"**, **"non-detailed text"**, and **"ambiguous data"** confuse you. **Fickle video** means that the video frame frequently changes in multiple scenes (e.g., **Figure. 6(a) in Appendix**). In our definition, a multi-scene video usually has richer semantics than a single-scene video. Therefore, it has high semantic relevance with more descriptions, and it is hard to summarize with a simple caption or word. **The non-detailed text** means text with general semantics, which can semantically match more videos or images than detailed text. Fickle videos and non-detailed texts jointly make up **ambiguous data**. Additionally, we are sorry for lacking the legend in Figure 1(b), which may cause confusion to you. Now, we have corrected it, which can be found at the **Figure 7 in the attched pdf file of Common Response**. If there are any other confusing wordings or further questions, feel free to let us know and we are more than happy to address them. All of these will be updated in the final paper.

---

> ### Author Response · Authors · 2023-08-20
> **Follow-up discussion**
>
> Thank you for your valuable feedback on our submission, particularly your suggestion to **train on noisy datasets**. The insightful suggestion enhance the quality of our work and better strengthen our claims. We hope that these improvements will be taken into consideration. If we fully address your concerns about our paper we would be grateful if you could re-evaluate our paper. If you have additional concerns, we remain open and would be more than happy to discuss with you.

---

> ### Comment · Area_Chair_Q568 · 2023-08-20
>
> Dear Reviewer tGbU,
>
> This is another friendly reminder to acknowledge that you have read the rebuttal and the other reviews. Please also share how they change your view on the paper, if at all. Thanks again for your service!
>
> Best,
>
> AC

---

> > ### Comment · Reviewer_tGbU · 2023-08-20
> >
> > This paper needs a major revision. I can't accept it in its current form.

---

> > > ### Author Response · Authors · 2023-08-21
> > > **Response to Reviewer tGbU**
> > >
> > > Thanks for your efforts in reviewing. We have carefully read your comments and provided detailed answers. Your comments about testing on noisy image-text pairs well encourage us to further consider the application of PAU. Now, we have revised several parts of our paper to make our claim clearer. If possible, we hope you can clarify your remain concerns or puzzles to further improve our paper quality. We are happy to address these concerns.

---

### Official Review · Reviewer_VBVJ · 2023-07-07

**Soundness:** 3 good
**Presentation:** 2 fair
**Contribution:** 2 fair
**Rating:** 6
**Confidence:** 5

**Summary:**

This paper proposes prototype-based aleatoric uncertainty quantification (PAU) to represent aleatoric uncertainty (i.e., data uncertainty) for cross-modal retrieval tasks. The proposed method uses the CLIP backbone, where the initial similarity matrix M is computed by the output of the CLIP encoders. Then, "evidence" $e$ is computed by the similarity between each modality embedding output and the prototype vectors. Using $e$, the belief $b$ and the uncertainty $u$ is computed. As far as the reviewer understood, the computed $u$ represents how the current embedding is *different from* the prototype vectors, i.e., if the current embedding is specifically different (in terms of the cosine similarity) from the prototype vectors, then its uncertainty becomes large. Using the learned uncertainty $u$, the proposed method re-rank the retrieved items by cosine similarity of the output of the CLIP encoders. Two losses are introduces to make the prototype vectors better, the uct loss (minimize the distance between average embedding vectors and prototypes) and the diversity loss (orthogonal regularization for prototype vectors). The effect of the proposed method is evaluated on four datasets.

**Strengths:**

- Measuring aleatoric uncertainty for multi-modal learning is an interesting and important problem.
- The proposed method looks efficient and easily adaptable to various methods.
- The proposed method is a somewhat straightforward extension of Subjective Logic (SL) [27] from the fixed class labels to the learnable prototypes. Hence, it may have the good theoretical properties shown by the SL paper [27], although it is not explicitly described in the paper.

**Weaknesses:**

My concerns can be summarized as:

- It is not clear that replacing fixed class information of SL with learnable prototype embeddings will ensure a proper uncertainty estimation.
- This paper misses the important previous work on the aleatoric uncertainty estimation method for cross-modal retrieval [A].
- The comparisons of MS-COCO 5K would not be fair (only the proposed method uses the pre-trained CLIP model), and many recent state-of-the-art methods are missing.

### Can the proposed method provide a proper uncertainty estimation?

The proposed method highly depends on SL [27], where SL uses fixed class information; hence, there is no learnable prototype. As this paper introduces new learnable prototypes, the proposed method requires two additional objective functions, Eq (5) and Eq (6). I am not sure how Eq (5) and Eq (6) will guarantee the quality of the proper uncertainty measure.

Eq (6) looks okay, but it enforces the orthogonality of the prototype embeddings, not the mutual exclusion among the embeddings. The latter one would be derived from independence, not orthogonality. Note that independence and orthogonality are highly related and somewhat exchangeable in many cases, I am not sure that smaller orthogonality regularization (i.e., Eq (6)) can ensure the weaker dependency between the prototype vectors. Moreover, I am not sure that Eq (6) is specifically better than widely-used orthogonality regularization, such as $\| I - Z^\top Z \|_F^2$, where $I$ denotes the identity matrix, and $Z$ denotes the full prototype vectors. Another possible alternative is using Maximum Mean Discrepancy (MMD) or Hilbert-Schmidt Independence Criterion (HSIC), whose values directly induce the strength of the dependency between two probabilistic densities. In my opinion, Eq (6) could be better, but Eq (6) itself is okay and can be somewhat sound regularization.

To me, the meaning of Eq (5) is ambiguous. It seems that Eq (5) behaves as minimizing the distances between prototypes and the embeddings; thereby, it enforces to make the value of $e_K$ larger (i.e., make the uncertainty value $u$ smaller). I am not sure how Eq (5) helps the proper uncertainty estimation.

I think both Eq (5) and (6) can be explained with some in-depth discussions with minimal changes in the manuscript, but as of now, I am not fully confident whether the proposed method can represent proper uncertainty estimations despite of using learnable prototypes rather than fixed class information.

### Missing related works on uncertainty estimation

This paper misses many directly related works on uncertainty estimation and cross-modal retrieval. Especially this paper did not consider PCME [A], a probabilistic embedding approach to represent aleatoric uncertainty (i.e., the quality of the data) for cross-modal retrieval.

- [A] Chun, Sanghyuk, et al. "Probabilistic embeddings for cross-modal retrieval." Proceedings of the IEEE/CVF Conference on Computer Vision and Pattern Recognition. 2021.

I think [A] should be discussed in the related work section and in the experiment section if possible.

There are also more aleatoric uncertainty estimation methods based on probabilistic embeddings, such as:

- Shi, Yichun, and Anil K. Jain. "Probabilistic face embeddings." Proceedings of the IEEE/CVF International Conference on Computer Vision. 2019.
- Park, Jungin, et al. "Probabilistic representations for video contrastive learning." Proceedings of the IEEE/CVF Conference on Computer Vision and Pattern Recognition. 2022.
- Neculai, Andrei, Yanbei Chen, and Zeynep Akata. "Probabilistic compositional embeddings for multimodal image retrieval." Proceedings of the IEEE/CVF Conference on Computer Vision and Pattern Recognition. 2022.
- Sun, Jennifer J., et al. "View-invariant probabilistic embedding for human pose." Computer Vision–ECCV 2020: 16th European Conference, Glasgow, UK, August 23–28, 2020, Proceedings, Part V 16. Springer International Publishing, 2020.

I would like to suggest adding discussions of the comparisons of the SL-based aleatoric uncertainty estimation and the probabilistic embedding-based aleatoric uncertainty estimation.

### Missing comparison methods for MS-COCO 5K

It could be a minor issue, but I think Table 4 misses many relevant and state-of-the-art methods for cross-modal retrieval. For example, as my previous comment, PCME [A] is a highly related work to this paper. In terms of using multiple diverse embeddings, [B] could be another relevant work to the proposed method, which is more relevant to PVSE [49]. I also would like to suggest adding more recent state-of-the-art cross-modal retrieval works, such as VSE$\infty$ [C].

- [B] Kim, Dongwon, Namyup Kim, and Suha Kwak. "Improving Cross-Modal Retrieval with Set of Diverse Embeddings." Proceedings of the IEEE/CVF Conference on Computer Vision and Pattern Recognition. 2023.
- [C] Chen, Jiacheng, et al. "Learning the best pooling strategy for visual semantic embedding." Proceedings of the IEEE/CVF conference on computer vision and pattern recognition. 2021.

Especially, Table 4 can be unfair because all other comparison methods do not use the pre-trained CLIP backbone; hence their performances are much worse than a simple "baseline" of this paper. Therefore, in my opinion, Table 4 should contain the comparisons with other methods using the pre-trained CLIP backbone.

However, I fully understand that it is not easy to compare with other cross-modal retrieval methods, especially when they apply different backbones, pre-extracted features, and optimization techniques. After a quick Google search, I found a very recent work, that uses the CLIP ViT-B/32 backbone for cross-modal retrieval:

- [D] Chun, Sanghyuk. "Improved Probabilistic Image-Text Representations." arXiv preprint arXiv:2305.18171 (2023).

I think the experimental results of [D] can be directly comparable with Table 4 of this paper, including VSE $\infty$, PCME, and InfoNCE. I would like to suggest adding the results of [D] to the main paper if the authors are willing to update their table to contain more recent papers. *Note that [D] is not a direct comparison method of the proposed method because it arxived after the NeurIPS submission deadline*; therefore, I don't argue that [D] is a significant related work that should be compared with the proposed method, but I bring [D] because [D] uses the same CLIP ViT-B/32 backbone with the experiments in this paper, and it reported [A] and [C] with the ViT-B/32 backbone, listed in my review.

### Minor comments

- The implementation details of the overall framework are somewhat hard to catch at first glance. I suggest adding the detailed process for computing $e$ and $u$ in Figure 2, and adding the details of $e$ (Table 7 in Appendix) to the main paper.

**Questions:**

Please check my comments in the weakness section. My major concerns are (1) I am not sure whether applying SL to learnable prototypes can lead to proper uncertainty estimations (2) missing significant related works, such as [A] and [B] ([B] could be considered as a contemporary work because [B] is a CVPR 2023 paper, that is published after the NeurIPS deadline).

**Limitations:**

I don't think this paper has a potential negative societal impact.

---

> ### Author Rebuttal · Authors · 2023-08-09
>
> We thank the reviewer VBVJ for the positive, patient, professional review and the valuable suggestions for improvement. Our responses to the raised questions are below:
>
> ***Q1 : Compared with probabilistic-based uncertainty estimation.***
>
> **A1:** Thanks for your professional and valuable comments. PCME [6] is a probabilistic-based approach to represent instance diversity, which can also represent uncertainty. The more diverse the instance, the more uncertain the instance. Thus, an instance's uncertainty and diversity should be positively correlated. To verify the superiority of SL-based method over than probabilistic-based method, we conduct two experiments, pearson correlation analysis and uncertain data analysis.
>
> (1) **Pearson correlation analysis**. The pearson correlation coefficient (**PCC**) $r_{xy}$ ($r_{xy}\in [-1,1]$) is a metric to estimate the strength of the correlation between variable $x$ and $y$. Generally, $r_{xy}>0.5$ means that two variables are strongly positively correlated. As mentioned, an instance's uncertainty and diversity should be positively correlated for PCME. A diverse instance will have a high mean similarity $h$ with the instances from the other modality. This means the uncertainty score $u$ and mean similarity $h$ should also be positively correlated. Thus, we compute the PCC between $u$ and $h$ to explore whether PAU correctly estimates uncertainty. The results are shown below.
>
> |Method|PCC-V|PCC-T|Dataset|
> |-|-|-|-|
> |PCME|-0.071|-0.219|MSCOCO|
> |PAU|0.886|0.786|MSCOCO|
> |PAU|0.917|0.939|MSR-VTT|
>
> In this Table, we compute the PCCs of PCME and PAU on MSCOCO, and PCCs of PAU on MSR-VTT. PCC-V and PCC-T separately indicate the instance's PCC in vision or textual modality. The low PCCs of PCME mean current probabilistic models have limited abilities to reflect uncertainty and diversity. We argue the reason arises from the strong prior distribution assumption, i.e., gaussian or other simple distributions are powerless to fit complex and diverse relationships in high-dimension space. By contrast, PAU obtains high PCCs in all settings, proving our approach indeed leads to proper uncertainty estimations.
>
> (2) **Uncertain data analysis**. To further prove the superiority of the SL-based method over the probabilistic-based method, we compare the changes of R@1 after removing top-r instances with the highest uncertainty scores quantified by PCME and PAU, respectively. To fairly compare, we employ removal on both predictions arising from CLIP and PCME. The results with the removed ratio from 0% to 20% on MS-COCO 5K is shown in **Fig. 8 of the pdf in common response**. We can observe that PAU outperforms PCME in all directions and predictions, meaning the uncertain data found by PAU is more precise than PCME. All experiments indicate the SL-based approach (PAU) works better than the probabilistic-based method (PCME) to represent aleatoric uncertainty.
>
> ***Q2 : Does PAU provide a proper uncertainty estimation?***
>
> **A2:** In A1, high PCCs of PAU mean the uncertainty score $u$ and mean similarity $h$ are strongly positively correlated, proving that our approach indeed leads to proper uncertainty estimations.
>
> Besides, in Fig. 3 of the main paper, we conduct experiments to explore the impact of uncertain data by comparing the retrieval results removing these uncertain data and removing random data. The results show that removing these uncertain data found by PAU can significantly improve retrieval performance. This can also prove that PAU possesses the ability to precisely estimate uncertainty.
>
> ***Q3 : The discussion of Eq. (6)?***
>
> **A3:** Thanks for your valuable comments. To verify the effectiveness of Eq. 6, we randomly sample K embeddings with Xavier init, representing the prototypes before training. Then we compute their mean similarity $\bar s$ between each other. Afterward, we compare it with the mean similarity between each prototype. We repeat the sampling trial 5 times and report the results below.
>
> ||$t_1$|$t_2$|$t_3$|$t_4$|$t_5$|$P_V$|$P_T$|
> |-|-|-|-|-|-|-|-|
> |$\bar s$|-1e-2|5e-3|-1e-2|-1e-2|-1e-2|9e-6|-2e-5|
>
> $P_V$ and $P_T$ denote the mean similarity of visual and textual prototypes, respectively. We can see that the mean similarity scales of $P_V$ and $P_T$ are about 0.001 to 0.0001 times the scale of random sampling, which means that Eq. 6 indeed pulls prototypes away from each other to a considerable distance.
>
> Besides, we also conduct the experiments on MSR-VTT using widely-used orthogonality regularization $|I-Z^\top Z|_F^2$ and Maximum Mean Discrepancy (MMD) instead of Eq. 6. The results are shown in Table below. We can observe that original Eq. 6 obtain the best performance on both t2v and v2t.
> |||t2v|||v2t||
> |-|-|-|-|-|-|-|
> |Method|R1|R5|R10| R1|R5|R10|
> |orthogonality|47.7|72.4|82.4|47.5|72.4|82.3|
> |MMD|47.6|72.2|81.8|47.6|72.0|82.1|
> |original Eq. 6|48.5|72.7|82.5|48.3|73.0|83.2|
>
> ***Q4 : The meaning of Eq. 5?***
>
> **A4:** In this paper, we define that an uncertain instance should have higher mean similarities with the instances from the other modality. i.e., as the A1(1) mentioned, the uncertainty score $u$ and mean similarity $h$ should also be positively correlated for an instance. Thus, Eq. 5 can be regarded as a constraint function to force $u$ closer to $h$. Then we can get a more precise uncertainty estimator.
>
> ***Q5 : Missing comparision with ViT-B/32 backbone-based methods on MS-COCO 5K.***
>
> **A5:** Thanks to the nice reviewer for providing the necessary papers. Limited by the Rebuttal character number, we discuss these in **"More Method Comparision in Different Noise Ratios" of Common Response**.
>
> ***Q6 : Adding some detailed processes for computing e and u in Fig. 2 and main paper.***
>
> **A6:** Thanks for your suggestion. We will add details of e, u computing process to Fig. 2. Meanwhile, we will transfer the e generation from Appendix to the main paper. Limited by the Rebuttal character number, we will update it in the final paper.

---

> > ### Comment · Reviewer_VBVJ · 2023-08-16
> >
> > Thanks to the authors for answering my questions. The response looks great to me.
> >
> > Especially, I think the additional experiments for comparing PCME and PAU in terms of uncertainty estimation will make the submission stronger. I am a bit surprised by the additional ablation study result for Eq. 6. It will be great if the authors can add a related discussion in the revised paper (e.g., what could be other options, why the proposed Eq 6 is better than others).
> >
> > Assuming that all the experiments and the discussions covered by the responses will be included in the revised paper, I will raise my score to "weak accept". Also, I strongly recommend including related works (e.g., aleatoric uncertainty estimation methods based on probabilistic embeddings, [A, B, C], and maybe [D]) listed in this comment.

---

> > > ### Author Response · Authors · 2023-08-17
> > >
> > > Thanks to the professional Reviewer VBVJ for the positive response and efforts to improve our work. The results of Eq. 6 also surprised us. We will conduct the experiments to explore the reason, test other methods such as Hilbert-Schmidt Independence Criterion (HSIC), and discuss them in the revised paper. All the supplemental experiments and discussion is helpful to improve our paper, we are willing to contain them in the revised paper. The related references you mentioned are really necessary for our paper, we will add all of them in the final paper.

---

### Official Review · Reviewer_qbVB · 2023-07-09

**Soundness:** 3 good
**Presentation:** 2 fair
**Contribution:** 2 fair
**Rating:** 5
**Confidence:** 4

**Summary:**

This paper presents the Prototype-based Aleatoric Uncertainty Quantifications (PAU) framework, which enhances the reliability of a pre-trained vision-language model (CLIP) by leveraging evidential knowledge. The paper introduces learnable prototypes as semantic categories and maps images and texts to these categories for uncertainty quantification. The proposed method improves model training and ranking in image-text retrieval tasks.

**Strengths:**

- The paper addresses the issue of model reliability and provides a reasonable approach to quantify data uncertainty in image-text retrieval tasks.
- The introduction of trainable prototypes complements contrastive learning-based vision-language alignment by incorporating evidence knowledge. This leads to improved reliability of the pre-trained model.
- The paper provides extensive experimental results and a detailed ablation study, demonstrating the effectiveness of the proposed method compared to existing image-text retrieval approaches.

**Weaknesses:**

- The loss function defined in Equation 5 raises confusion. The uncertainty is computed in the K category domain, while the mean similarity is computed in the N category domain. Could the author explain why these two domains are mapped together?
- The paper includes an ablation study on the number of trainable prototypes (K), showing that the best R@1 performance on MSR-VTT is achieved when K=8. Concerning the scalability of the method, would K vary on different datasets to achieve the best performance?
- Figure 5 visualizes the uncertainty of videos and captions. However, the connection between the data pairs is not clear. Could the authors explain the rationale behind selecting these specific examples?
- The paper lacks a clear introduction to the baseline models, including their training process and model structure. It would be helpful to provide more details in this regard. Also, the reporting of experimental results seems inconsistent or incomplete.

**Questions:**

Please refer to my question in the Weaknesses section.

---

> ### Author Rebuttal · Authors · 2023-08-09
>
> We thank the reviewer qbVB for the valuable and constructive feedback. We will address the raised questions point-by-point as below.
>
> ***Q1 : The loss function defined in Equation 5 raises confusion. Why uncertainty computed by K category domain and mean similarity computed by N category domain are mapped together?***
>
> **A1:** Thanks for the constructive question. In the Introduction, we explain why multi-scene videos and non-detailed texts have high uncertainty using information entropy. And in the histogram of Fig. 1(a), we can find that the multi-scene video B have a higher mean similarity with the text modality than Video A. Thus, we define that an instance with high uncertainty score $u$ should have high mean similarity $h$ with the instances from the other modality, i.e., $u$ and $h$ should be positively correlated. In the training process, given a specific video $q$, we use $K$ prototypes to compute $u_q$ and use $N$ captions to compute mean similarity $h_q$. According to our definition before, $u_q$ and $h_q$ should be positively correlated, and they have the same range $[0,1]$. Thus, we can adopt Eq. 5 to force $u_q$ closer to $h_q$ directly.
>
> ***Q2 : Would K vary on different datasets to achieve the best performance?***
>
> **A2:** To verify the scalability of the method, we additionally conduct K-varied experiments on MSVD and DiDeMo datasets. All results of K ablation are shown below.
> |||MSR|VTT||||||DiDeMo||||||MSVD|||
> |-|-|-|-|-|-|-|-|-|-|-|-|-|-|-|-|-|-|
> |||**t2v**|**v2t**||**Mean**|||**t2v**|**v2t**||**Mean**|||**t2v**|**v2t**||**Mean**|
> ||**R1**|**R5**|**R1**|**R5**|**R1**||**R1**|**R5**|**R1**|**R5**|**R1**||**R1**|**R5**|**R1**|**R5**|**R1**|
> |K=1|47.2|72.2|46.1|72.7|46.7||46.9|75.9|46.1|74.6|46.5||46.4|76.9|64.8|93.2|55.6|
> |K=4|47.6|72.8|46.7|71.6|47.2||48.0|75.8|47.3|74.8|47.65||46.6|77.0|67.9|92.9|57.25|
> |K=8|48.5|72.7|**48.3**|**73.0**|**48.4**||**48.6**|76.0|**48.1**|74.2|**48.35**||**47.3**|**77.4**|**68.9**|**93.1**|**58.1**|
> |K=12|**48.6**|72.4|46.9|72.6|47.8||48.1|75.8|46.2|75.4|47.15||46.9|77.1|68.3|92.2|57.6|
> |K=16|47.8|**73.0**|47.0|72.8|47.4||**48.6**|**76.9**|47.7|**75.5**|48.15||47.0|**77.4**|**68.9**|92.5|57.95|
>
> We find that the best Mean R@1 scores are obtained when K=8 in all three datasets. Meanwhile, when K is greater than 8, the performance change caused by K growth is minimal.
>
> ***Q3 : What is the rationale behind selecting the examples in Fig. 5?***
>
> **A3:** This paper mainly focuses on quantifying the uncertainty of a single video (image) or a single text, not vision-language pairs. Thus, the images and texts in Fig. 5 don't have a special connection. We want to show the samples with uncertainty scores sorted from large to small in text and video modalities, respectively. For example, the first row is the samples of a text and a video with high uncertainty. The third row is the samples of a text and a video with low uncertainty.
>
> If we want to compute the confidence $c$ of a vision-language pair, we can utilize the equation like $c=(1-u_v)\times(1-u_t)$. For example, assuming the text and the video in the first row of Fig. 5 make up a vision-language pair. The data quality confidence of this pair can be computed as $c=(1-0.984)\times(1-0.881)=0.001904$.
>
> ***Q4 : More details of the baseline training process and their model structure.***
>
> **A4:** For image-text retrieval, we utilize CLIP [1] as the baseline straightly. As for video-text retrieval, the training process and model structure can be found in **Fig. 9 of attached PDF file in common response**. We follow the model structure of the CLIP4CLIP [2] to generate $J$ frame representations $Z=\\{z^f_i|i=1,2,..,J\\}$ for each video and a text representation $z^t$ for each caption. Afterward, two branches are designed to separately catch global and local information. In the global branch, We first adopt a mean pooling to aggregate $J$ frame representations to get an "average global representation" $\bar z^v$. Next, the cosine similarity $s_{global}$ between $\bar z^v$ and $z^t$ is calculated. In the local branch, we first calculate a set of cosine similarities $S \in \mathbb R^J$ between $z^t$ and each frame representation $z^f_i$. Then we take the maximum value in $S$ as $s_{local}$. Finally, we take the average of $s_{global}$ and $s_{local}$ as the final similarity score, which will be constrained by a cross-entropy loss.
>
> ***Q5 : The reporting of experimental results seems inconsistent or incomplete.***
>
> **A5:** Thanks for your comment. We supplement several experiments. We first compare PAU with some CLIP ViT-B/32 backbone-based methods on MS-COCO with different noise ratios. The results are shown in Table below. We can observe that our methods outperform all other methods with the same backbone. Furthermore, PAU emerges a powerful ability for robust learning even in 50% noise ratio.
> |||AVG R1||
> |-|-|-|-|
> |**Method**|**0% noise**|**20% noise**|**50% noise**|
> |VSE$\infty$ [5]|**55.2**|51.4|18.4|
> |PCME [6]|53.0|48.1|43.0|
> |NCR [4]|-|48.8|45.5|
> |CLIP [1]|54.7|48.3|46.8|
> |PAU|**55.2**|**51.8**|**49.6**|
>
> Besides, We analyze the complexity of PAU and compare it with some famous previous methods. Concretely, We compute the FLOPs and parameter number of PAU on MSR-VTT. The results are shown in Table below. We find that the FLOPs and the parameter number of PAU are both significantly smaller than other methods, which means that PAU is a light-weight approach and easy to deploy.
> |Method|FLOPs (G)|Params (M)|
> |-|-|-|
> |VATT [10]|792|327|
> |Frozen [11]|771|180.9|
> |MILES [12]|771| 180.9|
> |PAU|**36**|**84.2**|
>
> We also conducted several other experiments, such as pearson correlation analysis to prove the prototypes correctly represent the overall semantics, uncertain data analysis to explore the superiority of PAU over probabilistic methods, etc. Limited by the rebuttal character number, we cannot show more experiment results. But we will add all these experiment results in the final paper.

---

> > ### Comment · Reviewer_qbVB · 2023-08-21
> > **Post after rebuttal**
> >
> > Thanks for providing more results. While the intuition behind pushing $u_q$ to $h_q$ remains unclear, the new experimental results partially address my previous concerns, and the method seems to consistently work well on different datasets. Thus, I tend to raise my score.

---

> > > ### Author Response · Authors · 2023-08-21
> > > **Response to Reviewer qbVB**
> > >
> > > Thanks to the reviewer qbVB for the positive response. As for $u_q$ and $h_q$, we can imagine that a multi-scene video can usually match multiple semantics, such as the video B in Fig. 1(a). It can match to 3 semantics of "talking", "shadow", and "cavern". By contrast, the single-scene video A in Fig.1 (a) approximately only match the semantics of "talking". Thus, the video B will have higher average similarity $h_q$ with textual modality than video A, because it not only have high similarities with the descriptions related to "talking", but also with the descriptions related to "shadow" and "cavern". That is to say, if a video matches more semantics, it becomes more uncertain, because it is harder to be summarized using a simple description. Therefore, the more uncertain the instance, the higher the average similarity with another modality. $u_q$ and $h_q$ should be positively correlated. This is the reason of pushing $u_q$ to $h_q$. Hope this can address your puzzle. All the additional experiments and discussion will be supplemented in our revised paper.

---

> ### Author Response · Authors · 2023-08-20
> **Follow-up discussion**
>
> Thank you for your valuable feedback on our submission, particularly your suggestions to **further explain the Eq. 5** and explore **K vary on different datasets**. These insightful suggestions enhance the quality of our work and better strengthen our claims. We hope that these improvements will be taken into consideration. If we fully address your concerns about our paper we would be grateful if you could re-evaluate our paper. If you have additional concerns, we remain open and would be more than happy to discuss with you.

---

> ### Comment · Area_Chair_Q568 · 2023-08-20
>
> Dear Reviewer qbVB,
>
> This is another friendly reminder to acknowledge that you have read the rebuttal and the other reviews. Please also share how they change your view on the paper, if at all. Thanks again for your service!
>
> Best,
>
> AC

---

### Author Rebuttal · Authors · 2023-08-09

# Common Response #
We appreciate all reviewers for their kind reviews and recognizing our work is helpful to multi-modal learning.

According to reviews, **highlights of this paper include**:

- The problem this paper addressed is interesting, novel, and important. (**VBVJ, tGbU, b4gq, hqQV**)
- The proposed method is reasonable, technically valid, and theoretically good. (**qbVB, VBVJ, tGbU, hqQV**)
- The experimental evaluations in the paper provide strong evidence to support our claims. (**qbVB, b4gq, hqQV**)
- The paper is well-written and easy to follow. (**hqQV**)

Reviewers **qbVB** ,**VBVJ**, **b4gq**, **hqQV** mark our work as good soundness. Furthermore, Reviewers **VBVJ**, **b4gq**, **hqQV** give the positive comments. They suggest acceptance.

**Major weakness of this paper** contains:
- Some confusion in uncertainty loss.
- Some points should be further discussed.
- Lack of some related works.
- Lack of some experiments.

Based on the above summary, we answer each question raised by Reviewers.

**Note**: **A pdf file** is attached in the common response to help answer some questions. It contains **all the new figures** we used in the rebuttal phase.

For some common questions, we conduct the experiment as follows:

## More Method Comparision in Different Noise Ratios ##

To make the comparison more fair on the image-text retrieval task. We follow the report format of [3], showing the average R@1 of i2t and t2i on several CLIP ViT-B/32 backbone-based methods on MS-COCO 5K, including VSE$\infty$ [5], PCME [6], NCR [4], and CLIP [1]. CLIP is our baseline. We can observe that PAU shows powerful performance.

Additionally, we explore the robustness of PAU training under noisy correspondences. We follow the noise split of NCR with 20%, 50% relationships randomly shuffled. It can be seen that PAU outperforms all other methods with a significant gap in 20% and 50% noise ratio, which shows that PAU also possesses strong robust learning ability.

|||AVG R1||
|-|-|-|-|
|**Method**|**0% noise**|**20% noise**|**50% noise**|
|VSE$\infty$ [5]|**55.2**|51.4|18.4|
|PCME [6]|53.0|48.1|43.0|
|NCR [4]|-|48.8|45.5|
|CLIP [1]|54.7|48.3|46.8|
|PAU|**55.2**|**51.8**|**49.6**|

All new experiments and supplemented contents will be updated to the final paper.

## Some References Mentioned in Our Responses. ##

[1] Radford, Alec  et. al. "Learning Transferable Visual Models From Natural Language Supervision." ICML, 2021.
[2] Luo, Huaishao  et. al. "CLIP4Clip: An Empirical Study of CLIP for End to End Video Clip Retrieval." Neurocomputing, 2022.
[3] Chun, Sanghyuk. "Improved Probabilistic Image-Text Representations." arXiv preprint arXiv:2305.18171.
[4] Huang, Zhenyu et. al. "Learning with noisy correspondence for cross-modal matching." NeurIPS, 2021.
[5] Chen, Jiacheng, et al. "Learning the best pooling strategy for visual semantic embedding." CVPR. 2021.
[6] Chun, Sanghyuk, et al. "Probabilistic embeddings for cross-modal retrieval." CVPR. 2021.
[7] He, Kaiming, et al. "Delving Deep into Rectifiers: Surpassing Human-Level Performance on ImageNet Classification." ICCV, 2015.
[8] Andrew, Saxe, et al. "Exact solutions to the nonlinear dynamics of learning in deep linear neural networks." ICLR, 2014.
[9] Glorot, Xavier, et al. "Understanding the difficulty of training deep feedforward neural networks." AISTATS, 2010.
[10] Hassan, Akbari, et al. "VATT: Transformers for Multimodal Self-Supervised Learning from Raw Video, Audio and Text." NeurIPS, 2021.
[11] Max, Bain, et al. "Frozen in time: A joint video and image encoder for
end-to-end retrieval." ICCV, 2021.
[12] Ge, Yuying, et al. "MILES: Visual BERT Pre-training with Injected Language Semantics." ECCV, 2022.

---

### Decision · Program_Chairs · 2023-09-21

**Decision:**

Accept (poster)

**Comment:**

The paper provides a novel and meaningful approach to the data uncertainty in image-text retrieval tasks through an extension of Subjective Logic, inheriting its strong theoretical properties. There were initial issues with the presentation (e.g. equations 5,6) as well as missing discussion and comparison against prior approaches and potentially unfair comparisons (MS-COCO experiments). The authors have effectively addressed the concerns during the rebuttal phase. As such, we recommend accepting the paper. We *strongly* urge the authors to include *all* the additional results and references in the final draft.